

**Estimation of Nighttime Aerosol Optical Depths Using**
**the Ground-based Microwave Radiometer**
Guanyu Liu[1], Jing Li[1*], Sheng Yue[1], Lulu Zhang[1], Chongzhao Zhang[1]
[1] Department of Atmospheric and Oceanic Sciences, School of Physics, Peking
University, Beijing, China
Corresponding authors: Jing Li (jing-li@pku.edu.cn)
**Abstract**
Aerosol optical depth (AOD) is a crucial parameter for understanding the impact of
aerosols on Earth's atmosphere and air quality. However, existing remote sensing
methods mostly rely on the shortwave spectrum, which does not allow measurements
at nighttime. In this study, we made a first attempt to retrieve AOD from
ground-based microwave radiometer (MWR) measurements. Brightness temperatures
(BT) at the K band (from 22.23 GHz to 30.00 GHz) and V band (from 51.25 GHz to
58.80 GHz) are trained against daytime spectral AOD from sun-photometer
measurements together with temperature profile using the random forest regression
(RFR) retrieval model, and the model is then used to retrieve nighttime AOD. The
algorithm demonstrates satisfactory performance, with strong agreements with lunar
AOD retrievals. The results also reveal a distinct day-night cycle of AOD, with
nighttime AOD typically higher than its daytime value. The physical basis of our
approach is verified using vertical temperature and humidity profiles from sounding
observation and simulation results from WRF-Chem as well as the MonoRTM. Our
study provides an effective and convenient approach to estimate nighttime aerosol
loading from surface, which has great potential in environmental monitoring and
climate forcing research.





# 1. Introduction

Aerosols have a significant impact on weather patterns and the Earth's climate (Huang et al., 2014; Li et al., 2022; Li et al., 2019; Riemer et al., 2019), offsetting about one-third of the warming effect by anthropogenic greenhouse gases and influence large-scale circulation (Huang et al., 2014; Li et al., 2022). However, accurately assessing their role in radiative forcing is a major challenge (Fan et al., 2016; Ghan et al., 2016; Seinfeld et al., 2016). Monitoring aerosol optical depth (AOD) is crucial for understanding aerosol impacts on climate and air quality, as it reflects the total amount of aerosols in the atmosphere optically (Visioni et al., 2023; Yang et al., 2020). As a result, there have been extensive efforts to measure AOD by various methods.

Remote sensing, either ground based or space borne, is an effective way to retrieve column AOD (Chaikovsky et al., 2020; Mhawish et al., 2017; Omar et al., 2013; Sinyuk et al., 2020). Other observations measure physicochemical properties of aerosols instead of optical properties like AOD (Kremser et al., 2016; Li et al., 2016b). Mainstream aerosol remote sensing techniques rely on aerosol scattering of shortwave radiation in the ultraviolet and/or visible spectrum, thus only daytime AOD can be obtained (Sayer et al., 2019; Sun et al., 2021). However, aerosols typically have day-night variability, due to factors such as different emission sources, boundary layer structure, etc (Arola et al., 2013; Cachorro et al., 2004; Cachorro et al., 2008; Guo et al., 2017). Aerosols at nighttime also have detectable impacts on the radiative balance, since they usually exert a warming effect in contrast to the cooling effect at daytime (Chen and Zhao, 2024; Colarco et al., 2014; Zhang et al., 2022), particularly in polar regions with the rapid change of AOD between daytime and nighttime (Chen and Zhao, 2024; Stenchikov et al., 2002; Wei et al., 2021). In special cases such as aerosols above the open oceans, they consistently exert a cooling influence in both shortwave and longwave, yet for dust aerosols, they potentially exert a warming effect





in longwave during both day and night (Adebiyi et al., 2023; Feng et al., 2022; Song
et al., 2022).
Remote sensing of aerosol properties at night is a challenging task. Lunar photometer
emerges during recent years as an effective and relative accurate nighttime AOD
retrieval technique, and has been widely used within the Aerosol Robotic Network
(AERONET) (Barreto et al., 2016). However, this method can only provide data at ~
halftime each month since it requires a relatively large amount of moon-reflected
solar radiation (Barreto et al., 2017; Berkoff et al., 2011). Compared with lunar
photometer method, the star photometry provides reliable nighttime AOD
measurements by leveraging stellar irradiance, eliminating lunar phase corrections,
with long-term datasets revealing diurnal aerosol dynamics (Pérez-Ramírez et al.,
2011; Pérez-Ramírez et al., 2016; Pérez-Ramírez et al., 2008; Pérez-Ramírez et al.,
2015). Arctic deployments and further development such as using a wide-field imager
enhance its adaptability in extreme environments and spatiotemporal resolution,
addressing gaps in traditional sun-photometer-based nocturnal monitoring (Ebr et al.,
2021; Ivanescu et al., 2021; Ivanescu and O'neill, 2023). Other researches take
advantage of urban light to retrieve nighttime AOD from space from multiple sensors
(Jiang et al., 2022; Meng et al., 2023; Wang et al., 2023; Wang et al., 2020; Zhou et
al., 2021). For example, Zhang et al. examined the effectiveness of retrieving
nighttime AOD over urban areas by utilizing city lights observed through the VIIRS
(Visible Infrared Imaging Radiometer Suite) Day-Night Band (DNB) (Zhang et al.,
2019). However, this approach has limitations as it does not account for multiple
scattering and gas absorption, which can potentially reduce the signals from aerosols
(Zhou et al., 2021). Furthermore, these studies are constrained to the spatial scale of
urban areas, resulting in vast rural regions being unexplored (Meng et al., 2023).
Active remote sensing, such as lidars, can provide aerosol measurements at both day
and night time (Balmes et al., 2021; Jiang et al., 2024). Nonetheless, solving the lidar
equation requires assumption of the lidar ratio, and this assumed lidar ratio often



causes large uncertainty of the retrieved extinction profiles as well as column
integrated AOD usually (Liu et al., 2018; Rogers et al., 2014; Santa Maria and Winker,
2005). For the day-night difference of AOD, previous studies find slight increases of
nighttime AOD using the long-term sun-and-star photometry data (Pérez-Ramírez et
al., 2012; Pérez-Ramírez et al., 2016; Wang et al., 2004). Moreover, using Infrared
Atmospheric Sounder Interferometer (IASI) and Cloud-Aerosol Transport System
(CATS) are also effective methods to understand day-night differences in dust
aerosols (Tindan et al., 2023; Yu et al., 2021). However, existing research regarding
day-night difference of AOD only focuses on special types of aerosols such as dust
aerosols, and has low availability due to the moon phase and urban light extent
(Barreto et al., 2017; Jiang et al., 2022; Meng et al., 2023; Wang et al., 2023; Wang et
al., 2020; Zhou et al., 2021). Due to our limited capability to measure nighttime AOD,
there is a significant knowledge gap between daytime and nighttime aerosol
properties.
In contrast to shortwave radiation which is only available during daytime, longwave
radiation, especially in the thermal infrared and microwave spectrum, exists during
both day and night, and offers the potential to derive nighttime aerosol property
(Dufresne et al., 2002; Panicker et al., 2008). Previous research has explored the
possibility to retrieve aerosol loading using longwave measurements, but mostly
focused on large particles such as dust (Clarisse et al., 2019; Desouza-Machado et al.,
2010; Klüser et al., 2012; Pierangelo et al., 2004; Pierangelo et al., 2005; Zheng et al.,
2022; Zheng et al., 2023). For example, using collocated thermal infrared
observations from MODIS and dust optical depth from Cloud-Aerosol Lidar with
Orthogonal Polarization (CALIOP), Zheng et al. simultaneously retrieve the thermal
infrared dust optical depth and coarse-mode effective diameter over global oceans
(Zheng et al., 2023). Observational and simulation studies indicate that the microwave
brightness temperatures (BTs) and brightness temperature polarization differences
may be both useful for estimating the dust mass loading (Ge et al., 2008; Hong et al.,

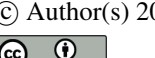



2008; Huang et al., 2007; Mitra et al., 2013). Our previous study utilized
satellite-based thermal infrared measurements in the atmospheric window region to
retrieve nighttime AOD (Liu et al., 2024), and proves the effectiveness of these
longwave measurements in deriving aerosol properties.
Ground-based microwave radiometer (MWR) is a widely used remote sensing
instrument to retrieve temperature and humidity profiles using emitted longwave
radiation by the surface-atmosphere system (Bianco et al., 2005; Greenwald et al.,
2018; Knupp et al., 2009). Considering the aforementioned concepts of utilizing
longwave radiances to retrieve aerosol properties and the potential alterations in
microwave BTs due to the modified temperature and humidity profiles resulting from
the shortwave radiation effect of aerosols, there is potential that aerosol information
can be derived from MWR measurements. Therefore, in this study, we explore the
possibility to retrieve AOD using surface based MWR measurements in the K spectral
bands (22.23 GHz, 22.50 GHz, 23.03 GHz, 23.83 GHz, 25.00 GHz, 26.23 GHz, 28.00
GHz, and 30.00 GHz) and V spectral bands (51.25 GHz, 51.76 GHz, 52.28 GHz,
52.80 GHz, 53.34 GHz, 53.85 GHz, 54.40 GHz, 54.94 GHz, 55.50 GHz, 56.02 GHz,
56.66 GHz, 57.29 GHz, 57.96 GHz, and 58.80 GHz). A machine learning based
algorithm is developed to estimate AOD during both day and night. The theoretical
basis of the method is further verified using regional model and radiative transfer
simulations. The difference between day and night time AOD is also examined using
the retrieval results.

## 2. Data and Methods

The retrieval algorithm used in this study is described in Figure 1 and includes four
main steps: (1) preprocessing of input variables, (2) training the Random Forest
Regression (RFR) retrieval model, (3) estimation of AOD using the trained model,
and (4) independent validation to refine the model and assess its performance



compared to lunar photometer observations. The details of the datasets and methods
are explained below.
**2.1 Datasets**
In this study, we utilized BT data collected from the MP-3000A MWR, which was
stationed at the Beijing Nanjiao Meteorological Observatory located in China
(39.80°N, 116.47°E). The MP-3000A MWR is capable of detecting signals in the
K-band (22 to 30 GHz) and V-band (51 to 59 GHz), and it is also equipped with
additional features such as a precipitation sensor, an infrared radiation thermometer,
and other relevant instruments. To maintain the accuracy and consistency of the
atmospheric BT measurements, the MWR undergoes regular real-time calibration.
These measurements are essential for obtaining temperature profiles and AOD data.
Our analysis focuses on the K and V band of BT observations with 22 available
channels, because BT observations at the K band are sensitive to water vapor
absorption and BT observations at the V band are sensitive to oxygen absorption and
temperature changes. We use the data ranging from December 2019 to October 2020
with a temporal resolution of one minute.
The measured BTs include inaccuracies and unusual values caused by instrumental
faults, calibration problems, and environmental factors. Hence, it's crucial to conduct
quality control (QC) checks on the BT data before processing it further. These checks
involve removing abnormal values to ensure that the BTs fall within a reasonable
temperature range of 2.7 to 330 Kelvin, and inspecting for data consistency over time
as per the methodology of Zhang et al. (Zhang, 2024). Notably, because the
collocation between MWR and Level 2 sun photometer AOD products from the
AERONET is already clear-sky data, there is no need to perform cloud screening on
the MWR data.



AOD retrieved using the solar and lunar methods at the Beijing-CAMS AERONET
site, which is the closest site to the MWR location, is used as training and validation
data in the retrieval algorithm. For training our model, we utilized Level 2 sun
photometer AOD products at the wavelengths of 440nm, 675nm, 870nm, and 1020nm
during the day. Version 3 Level 1.5 lunar AOD products at the same wavelengths to
validate AOD retrievals at night.
Given that MWRs are instrumental in tracking atmospheric temperature and humidity
profile changes (Zhang et al., 2024), our method retrieves vertical temperature
profiles concurrently. This is achieved by using temperatures at different pressure
levels obtained from the European Center for Medium-Range Weather Forecasts
(ECMWF) Reanalysis version 5 (ERA-5) as the target for our training. To further
assess the accuracy of the model in predicting vertical temperature profiles, we
utilized the collocated sounding data obtained from Beijing Meteorological Station
(station ID: 54511) during the corresponding time frame. These sounding data were
collected twice daily respectively at 0000 and 1200 UTC from December 2019 to
October 2020. For the physical interpretation of our retrieval method, we employed
collocated vertical profiles of temperature and relative humidity (RH) from the same
sounding data under varying aerosol loadings to explore the effects of aerosol loading
on the vertical profiles of meteorological variables. These vertical profiles were
further utilized to compute BTs using the monochromatic radiative transfer model
(MonoRTM).
**2.2 Retrieval Algorithm**
Because the relationship between aerosol loading and microwave radiation is
complicated and could be nonlinear, we use a machine learning based retrieval
method focusing on the RFR method (Svetnik et al., 2003). All variables are
appropriately matched in both space and time. Specifically, AOD from sun
photometer measurements and BTs from the MWR are matched within a 5-minute





time window, while hourly temperature profiles from ERA-5 reanalysis datasets and
BTs from the MWR are collocated within a 30-minute time window and a 15 km
spatial radius.
We first apply the relative importance feature selection technique, which is based on
the Gini importance measure (Nembrini et al., 2018), to identify significant
independent variables and build a generalized model. The relative importance of each
factor is presented in Figure 2. It is observed that BTs across various frequency bands
carry similar levels of importance, suggesting that the BTs are almost equally
important for retrieving AOD.
Subsequently, the retrieval algorithm is trained using eight selected K-band BTs as
input and target variables, which include sun photometer AOD at 440nm, 675nm,
870nm, and 1020nm, as well as ERA-5 vertical temperature profiles at 100 hPa, 200
hPa, 500 hPa, 700 hPa, 850 hPa, and 1000 hPa. To ensure the representativeness of
the sampling, we select the first 3/4 of the data in each month as the training set and
the last 1/4 of the data as the testing set. Additionally, the algorithm is adapted to
estimate nighttime AOD using nighttime BTs from microwave radiometry as inputs,
which is then validated against nighttime AOD observations from lunar measurements
in lunar photometer for the same period. Moreover, AOD, whether in the visible or
microwave region, is associated with aerosol loading, which serves as the foundation
for retrieving visible AOD using microwave observations. Since we primarily aim at
retrieving AOD rather than aerosol type, we did not consider AOD at the other
wavelengths when building the AOD retrieval model. The relationship between AOD
at 550nm and that at the microwave band is enclosed in the random forest model. The
model performance is assessed against photometer retrievals using metrics such as
linear regression slope and intercept, correlation coefficient (R), root-mean-square
error (RMSE), and mean absolute percentage error (MAPE).



The RFR model is built by varying the number of decision trees from 8 to 256.
Through validation analysis, it is determined that the optimal number of trees is 128,
based on the best performance during validation. After refining the algorithm through
extensive training and testing, it is used to retrieve nighttime AOD from nighttime
MWR BTs, with validation against collocated lunar AOD measurements from the
lunar photometer. Moreover, before investigating the diurnal cycle of MWR derived
AOD, we perform a quality control on the minute-resolution retrieval results that
typically have a higher noise level. Specifically, for each specific minute, we extract
the AOD for this minute from each day to form an AOD sequence. We then calculate
the mean and standard deviation of this AOD sequence. Finally, we remove AOD that
exceeds three times the standard deviation. Considering the suitable quantity of
outliers procured by setting the threshold at three standard deviations and the
prevalently utilized 3-sigma rule, we used three standard deviations as the threshold
(Li et al., 2016a; Liu et al., 2024; Wang et al., 2012).
**2.3 WRF-Chem simulations**
To investigate the effect of aerosols on downward microwave radiation, we use the
Weather Research and Forecasting model with Chemistry (WRF-Chem) simulations
combined with the MonoRTM radiative transfer model. Because MWR-observed BT
change is not only due to AOD change but also reflects the change of meteorological
conditions due to the AOD change, we apply WRF-Chem and MonoRTM radiation
transfer model instead of radiative transfer simulations only.
WRF-Chem simulation runs from 0000 UTC on 17 December 2016 to 0000 UTC on
20 December 2016 (a 72-hour period). The simulation period is different from that of
the retrieval because there are no updated emission fields for 2019 and 2020. The
initial meteorological conditions used for the simulations are based on the National
Center for Atmospheric Research (NCEP) Final Global Forecast System Operational
Analysis (FNL) provided by the National Oceanic and Atmospheric Administration





(NOAA), with a 1° × 1° spatial resolution and a 6-hour temporal interval. The
emission fields used here are Emissions Database for Global Atmospheric Research
(EDGAR), MIX, and Multi-resolution Emission Inventory for China (MEIC) (Crippa
et al., 2018; Li et al., 2017; Wang et al., 2014). The surface emissivity we used for
simulation is the default data for WRF-Chem. The simulation domain encompasses
the area of Beijing, Tianjin, and Hebei province (as shown in Figure 3), with a center
point at 40.00°N, 116.25°E. The model employs a three-tiered nesting configuration,
featuring outer grids of 40 × 46 with a 90 km horizontal spacing, middle grids of 48 ×
60 with a 30 km horizontal spacing, and inner grids of 51 × 72 with a 10 km
horizontal spacing. The vertical atmosphere is segmented into 47 levels, ranging from
the model's ground level to 100 hPa, encompassing both the surface and the upper
atmosphere. Figure 3 illustrates the domains of the WRF model simulations and the
location of the MWR deployed at the Beijing Nanjiao Meteorological Observatory in
China. To further confirm our findings, we perform another set of parallel
experiments lasting from 0000 UTC on 3 December 2016 to 0000 UTC on 5
December 2016 (a 48-hour period) with the same settings. The first day of both sets of
experiments is used for model stabilization, and the subsequent days are utilized for
analysis.
For the choices of physical parameterization schemes, we employ the Lin
microphysics scheme, the rapid radiative transfer model for global climate model
(GCM) applications (RRTMG) for shortwave radiation, the Yonsei University (YSU)
boundary layer scheme, the Monin-Obukhov ground layer scheme, the Carbon-Bond
Mechanism version Z (CBM-Z) for gas-phase chemistry, and the Model for
Simulating Aerosol Interactions and Chemistry (MOSAIC). The model output has a
one-hour temporal resolution.
To investigate the responses of surface downward microwave radiation to aerosol
loadings, we also conducted two parallel experiments with and without aerosol



267 emissions in the study. Two simulations that are respectively, designated as

268 "EXP_AER" and "EXP_NOAER" are carried out. The EXP_AER experiment is

269 defined as a control simulation in which aerosol and aerosol precursor emission

270 scheme is turned on. This aerosol emission includes emissions of carbon monoxide,

271 nitrogen oxides, sulphate oxides, dust aerosols, biomass aerosols, biomass burning

272 aerosols, sea salt aerosols and anthropogenic aerosols. The sensitivity experiment

273 ("EXP_NOAER") is also conducted by closing corresponding aerosol and aerosol

274 precursor emission scheme. The difference between control and sensitivity results are

275 considered as the adjustments of vertical meteorological profiles to aerosol loadings.

276 This method is also widely used to explore the radiative forcing of different kinds of

277 aerosol and its effects on meteorological fields in previous studies (Chen et al., 2023;

278 Matsui et al., 2018).

279 It is important to note that the aerosol-radiation interaction feature is activated in the

280 WRF-Chem model to investigate the impact of aerosol loadings on meteorological

281 fields. Subsequently, we input meteorological profile data from pollution cases

282 without cloud cover at each grid point into the monochromatic radiative transfer

283 model (MonoRTM) to calculate the corresponding BT responses at various

284 frequencies within the K-band.

285 **2.4 MonoRTM**

286 The MonoRTM, developed by Atmospheric and Environmental Research (AER), is a

287 radiative transfer model specifically designed for microwave and millimeter-wave

288 applications (Clough et al., 2005). This model is particularly useful in the microwave

289 radiation calculation (Payne et al., 2011). In this study, it is used to calculate the

290 brightness temperatures (BTs) associated with the simulated temperature and

291 humidity vertical profiles from WRF-Chem.



## 3. Results

### 3.1 Model fitting and validation

The retrieval model has great fitting performance, as shown by Figure 4. The model fitting reaches correlation coefficients of 0.98 for the 440 nm, 675 nm, 870 nm, and 1020 nm, respectively, albeit with a minor systematic low bias for high AOD scenarios, which is similar to MODIS AOD products (Levy et al., 2013). Due to the consistent model performance in all wavelengths (Figure 4), we will focus on results at 440 nm in the following discussions.

Figure 5 displays the comparison between the daytime and nighttime AOD independently retrieved by MWR using our algorithm and those from the sun and lunar photometer from December 2019 to October 2020. The model, tested during the daytime, utilized a dataset of over 3,000 samples and achieved correlation coefficients of 0.96 for 440 nm (Figure 5a). Most points are concentrated on the 1:1 line, with RMSE within 0.11 and MAPE within 0.11. The accuracy of this estimation is similar to existing shortwave-based algorithms (Levy et al., 2013). However, the key advantage of using microwave BT is the capability to retrieve AOD at night, a feature lacking in these shortwave-based algorithms (Figure 5b). Nighttime AOD retrieval reaches comparable performance to that for daytime, exhibiting a high correlation of 0.91 with lunar AOD. A minor systematic bias towards lower values in high AOD scenarios is also noted, with RMSE about 0.14 and MAPE approximately 0.28, indicating the overall satisfactory performance of MWR retrievals. In addition, the MWR results also well capture the spectral variation of AOD for fine (440 nm to 870 nm Angstrom index > 1) and coarse mode particles (440 nm to 870 nm Angstrom index < 1), as shown in Figure 6.

Our algorithm simultaneously retrieves daytime and nighttime temperature profiles. As shown in Figure 7 & Figure 8, atmospheric temperature retrieval results also



demonstrate good performance and exceed those of AOD. This is expected since the
main signals in the microwave come from emitted radiation by the atmosphere that is
directly related to temperature. In detail, R is generally above 0.98 and all of the
RMSEs are around 1.0 K in the training set (Figure 7). Similarly, for the test set, R is
above 0.95 and all of the RMSEs are around 1.8 K the test set (Figure 8), comparable
to previous studies using MWR data with an optimal estimation method (Cimini et al.,
2006). The significant biases at some pressure levels may be attributed to the larger
biases between sounding data and reanalysis data that is used to train the model
(Varga and Breuer, 2022). Our model also well captures the characteristics of the
climatological mean temperature vertical profile, with the error in each pressure layer
within 1.5 K (Figure 9a). There exist greater RMSE and bias in low pressure levels
partially due to the higher temperature variations in these levels, the overall RMSE
and bias serve to illustrate the exemplary performance of the model in estimating the
vertical temperature profiles (Figure 9b & c).
In summary, the day and nighttime MWR-based AOD and vertical temperature
profiles derived from our algorithm successfully capture the AOD variability and
vertical temperature profile characteristics with satisfactory accuracy. This model also
unveils the spectral characteristics of AOD, with higher wavelengths corresponding to
lower AOD. With great performance through model validation, we will investigate the
diurnal cycle of AOD in the following section.
**3.2 The diurnal cycle of MWR derived AOD**
We further examine the day-night differences in the AOD retrieved by MWR and
compare them to those revealed by surface photometer.
Figure 10a-b illustrates the mean diurnal cycles of the photometer AOD and
MWR-based AOD derived from BT observations at the Beijing Nanjiao
Meteorological Observatory in China. As shown in Figure 10a, mean diurnal AOD





follows a bi-modal temporal distribution, with a greater peak ~21:00 and a secondary
peak at ~03:00. The AOD stays relatively low from 06:00 to 10:00, gradually rises
from 10:00 to 21:00, reaching the first peak at 21:00. After that greater peak, the AOD
decreases from 22:00 to 00:00, and then increases again until it reaches the second
peak at 03:00. This pattern is consistent across other spectral bands (675 nm, 870 nm,
and 1020 nm, not shown here). Because the number of nighttime AODs from the
photometer is smaller than that during the daytime, but the number of nighttime
AODs from the MWR is nearly equal to that during the daytime, this decrease may
not be entirely explained by the lack of data sampling and needs further investigation
in the future study. Moreover, although the MWR-based AOD seems to underestimate
the extreme pollutions with high AOD compared with photometer observation, since
the number of upper outliers of AOD of the photometer is higher than that of MWR,
the overall temporal pattern is similar to that of the photometer (Figure 10a).
The mean and median AOD values further support the above findings, highlighting
higher nighttime AOD compared to daytime (Figure 10b). This difference is validated
by the boxplots of MWR-based AOD and photometer AOD (Figure 10c), passing the
Student's $t$-test significance test with $p \leq 0.05$. Specifically, the median daytime
AOD is in the range of 0.15 to 0.28 for MWR and 0.15 to 0.27 for the photometer,
while the median nighttime AOD is greater than 0.34 for MWR and higher than 0.30
for the photometer. Similarly, the mean daytime AOD is in the range of 0.25 to 0.35
for MWR and 0.24 to 0.32 for the photometer, while the mean nighttime AOD is
greater than 0.40 for MWR and over 0.44 for the photometer. This discrepancy
between daytime and nighttime AOD has also been observed in previous studies
estimating nighttime AOD by incorporating infrared radiance measurement from
AIRS into the machine learning model (Liu et al., 2024). Notably, the mean AOD
tends to exceed the median AOD, partly due to the long-tail distribution of AOD and
the presence of high extreme values (Sayer et al., 2019). Moreover, AOD at the other
wavelengths (675 nm, 870 nm, and 1020 nm) exhibit similar diurnal patterns with





peaks at about 20:00-22:00 (not shown here) and higher nighttime AOD in general
(Figure 6).
The increase in nighttime AOD compared to daytime can be attributed to various
factors, including a shallower mixed layer due to reduced horizontal mixing and
transport, a decrease in atmospheric environmental capacity, higher relative humidity,
enhanced aerosol hygroscopic growth, or intensified pollution emissions (Brock et al.,
2016). Similar observations of elevated nighttime particle matter concentration have
been reported in previous studies (Perrone et al., 2022; Su et al., 2023). However,
research on nighttime aerosol properties is limited, warranting further analysis to fully
understand these discrepancies, which exceeds the scope of this study.
In summary, by using the BT measured by the MWR to retrieve AOD during
nighttime, we can uncover the daily cycle of AOD. This improves our understanding
of the day-nighttime AOD variability, provides insights into the diurnal changes of
atmospheric pollution and sheds light on nighttime aerosol radiative effects.
**3.3 Physical interpretation**
Since the machine learning technique does not necessarily represent the physical
relationship between aerosol loading and microwave radiances, we further verify the
theoretical basis of our technique by analyzing the observed temperature and RH
profiles under various AOD levels and using WRF-Chem combined with MonoRTM
simulations. A set of sensitivity experiments with and without aerosol forcing is
conducted using WRF-Chem as described in Section 2, whose atmospheric profiles,
including temperature, water vapor, gases and aerosols, are then used as the inputs to
the MonoRTM to simulate the downward microwave radiances (represented by BT)
observed by the MWR. To mitigate the influence of surface temperature on BT, we
maintained a consistent surface temperature range (265 K-270 K) throughout the
simulation.

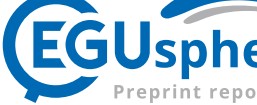

We first analyze the temperature and RH profiles from sounding observations under
various AOD levels (Figure 11a-b & d-e). These AOD levels include light pollution
(AOD<0.2), medium pollution (0.2<AOD<0.5), and heavy pollution (AOD>0.5)
scenarios. The selection of this threshold is to ensure a balanced sample size for each
scenario. All differences in the temperature and RH profiles under different AOD
scenarios passed the significance test with $p \leqslant 0.1$ by the $t$-test. For the temperature
profiles, a higher AOD corresponds to a lower temperature in the upper atmosphere,
and vice versa (Figure 11a). However, for the low-level atmosphere, the temperature
might first increase as AOD increases and then decrease with AOD as increases. This
is associated with aerosol type and optical properties (Che et al., 2024; Mahowald et
al., 2011). For the RH vertical profiles, RH increases as AOD increases at all pressure
levels (Figure 11b). This may be attributed to aerosol hygroscopic growth effect,
leading to a higher AOD (Quan et al., 2018). Notably, since the collocation between
MWR and Level 2 sun photometer AOD products from the AERONET is already
clear-sky data, the vertical profiles of RH is relatively low. BTs at 22.23 GHz
calculated by these vertical profiles from MonoRTM also demonstrate that BTs tend
to increase with AOD (Figure 11c). BTs at other frequencies in the K band also show
similar trend (not shown here). Similarly, the WRF-Chem output also demonstrates
the sensitiveness of temperature and RH vertical profiles to aerosol loading,
contributing to statistically significant BT difference under different pollution levels
(Figure 11d-f). The above observational evidence indicates that MWR estimate AOD
by detecting the temperature and humidity profile differences caused by the presence
of aerosols, further verifying the theoretical basis of our technique.
Furthermore, our simulation results, illustrated in Figure 12 and 13, indicate that for
all frequencies in the K band, BT increases as AOD levels increase. This phenomenon
exists in both the daytime and nighttime. Specifically, at 22.23 GHz, BT levels for
clean conditions range from 60 K to 80 K, while for polluted conditions they range
from 80 to 130 K, showing a statistically significant difference at both daytime and



nighttime (Figure 12a & 13a). BT levels at other frequencies support this trend,
indicating that BT tends to increase with AOD (Figure 12b-d & 13b-d). The increase
of K band BT with AOD might be related to coherent changes of water vapor and
aerosols, either due to aerosol absorption of water or meteorological conditions that
affect both water vapor and aerosols. In contrast to the observations in the K band, an
analysis of the V band frequencies reveals a consistent decrease in BT with the
reduction of AOD levels, applicable to both diurnal and nocturnal periods (Figure
12e-h & 13e-h), which well corresponds to the cooling effect of aerosols. Notably, at
a frequency of 51.76 GHz, the BT levels exhibit a range of 264 K to 270 K under
pristine atmospheric conditions, whereas under polluted conditions, these levels are
observed to be between 262 K and 265 K. Although the magnitude of this change is
less pronounced than that observed in the K band, it still passes the statistical
significance ($p \leqslant 0.1$ by the $t$-test), indicating a reliable and measurable effect. The
above-mentioned conclusion was further verified by simulations lasting from 0000
UTC on 3 December 2016 to 0000 UTC on 5 December 2016 (a 48-hour period) with
the same settings (not shown).
To deepen our understanding of the impact of aerosol loading on longwave radiation,
we conducted a comparative analysis using WRF-Chem. By comparing scenarios with
aerosol loadings (EXP_AER) and without aerosol loadings (EXP_NOAER), we
examined the differences in AOD, surface temperature (ST) and ground downward
longwave radiation (GDLR). The findings reveal that higher aerosol concentration
levels have a negative effect on ST (Figure 14b & e), particularly during the daytime
(Figure 14b), while positively influencing GDLR (Figure 14c & f), especially at
nighttime (Figure 14f), which is consistent with the above MonoRTM calculations.



## 4. Conclusions and Discussions

This study introduces a new method for estimating clear sky AOD using BT measurements in the K and V band obtained from the MWR. By establishing a strong correlation between the photometer AOD and multiple BTs derived from the MWR at the Beijing Nanjiao Meteorological Observatory using a machine learning algorithm, we were able to accurately retrieve nighttime AOD and vertical temperature profiles. This model also well captures the spectral characteristics of AOD with higher Angstrom index for fine-mode dominated AOD and lower Angstrom index for coarse-mode dominated AOD. After applying this model with satisfactory performance, we show that the AOD diurnal cycle and find that AOD values follow a bi-modal diurnal cycle temporal distribution, with a greater peak ~21:00 and a secondary peak at ~03:00, suggesting higher nighttime AOD compared with daytime. The difference between daytime and nighttime AOD observed in the MWR data well agrees with sun and lunar photometer observation as well as particle matter concentration observations.

The theoretical basis of our algorithm is also confirmed by analyzing observational vertical profiles of temperature and RH under various AOD levels and WRF-Chem as well as MonoRTM simulations. Observation indicated that the vertical profiles of temperature and RH have statistically significant differences ($p \leqslant 0.1$) under different AOD levels, suggesting that MWR might estimate AOD by detecting the temperature and humidity profile differences caused by various aerosol loadings. Simulation further indicated a consistent and mostly linear increase in BTs in the K band and decrease in BTs in the V band with AOD (550 nm) across all time periods. Aerosols tend to induce a cooling effect at surface while increasing ground downward longwave radiation, especially at the nighttime.

This study holds significant promise for environmental and climate research as MWR BT measurements can be obtained day and night without being hindered by bright



surfaces. The methodology developed here can potentially be applied to MWRs in
other locations worldwide to retrieve both daytime and nighttime AOD values.
However, it is important to note that this investigation is preliminary and may contain
uncertainties. It is also applicable under clear sky since during cloudy sky, the
downward microwave radiation will be dominated by that emitted by clouds.
Moving forward, we aim to explore additional aerosol characteristics that may be
inferred from BT measurements, such as aerosol absorption and layer height. This
will enhance our understanding of aerosol distribution and properties, ultimately
improving our ability to monitor and predict aerosol impacts on climate and the
environment.



## Code and data availability

The sun photometer AOD data was obtained from https://aeronet.gsfc.nasa.gov/new_web/webtool_aod_v3.html, last access: 20 Apr 2024; the lunar photometer AOD data was obtained from https://aeronet.gsfc.nasa.gov/new_web/webtool_aod_v3_lunar.html, last access: 20 Apr 2024; the temperature profile from the ERA-5 reanalysis data was downloaded from https://cds.climate.copernicus.eu/cdsapp#!/dataset/reanalysis-era5-pressure-levels?tab=overview, last access: 24 Apr 2024; the MonoRTM source code is available on https://github.com/AER-RC/monoRTM, last access 18 Apr 2024. The sounding data obtained from Beijing Meteorological Station (station ID: 54511) was obtained from https://weather.uwyo.edu/upperair/bufrraob.shtml.

## Author contributions

GL and JL conceived the study and wrote the original draft. GL, SY, LZ, and CZ ran the simulation and conducted the corresponding analysis. All authors revised and reviewed the draft.

## Competing interests

The authors declare that they have no known competing financial interests or personal relationships that could have appeared to influence the work reported in this paper.

## Acknowledgments

The authors thank Pawan Gupta and Elena Lind for their effort in establishing and maintaining Beijing-CAMS AERONET site.

## Financial support



This study is funded by the National Natural Science Foundation of China (NSFC) No.
42425503 and No. 42175144, and National Key Research and Development Program
of China (grant no. 2023YFF0805401).



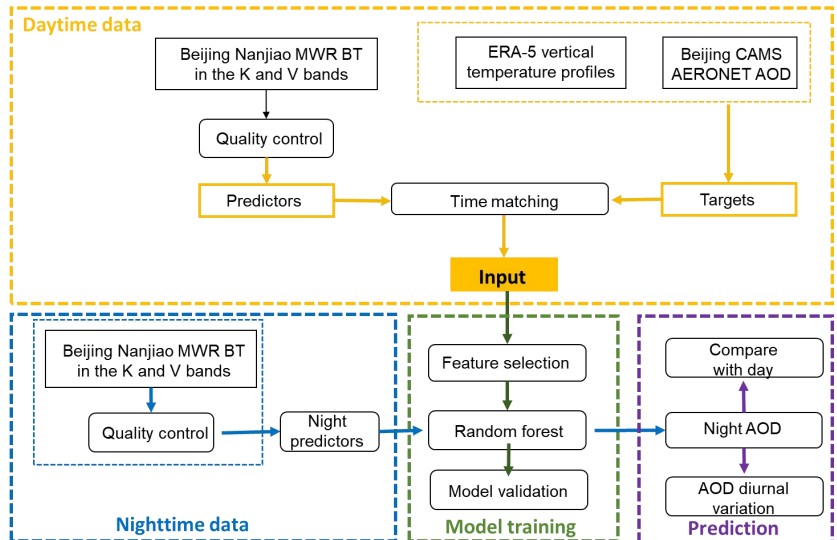


**Figure 1.** The flowchart for clear sky nighttime AOD retrieval algorithm.




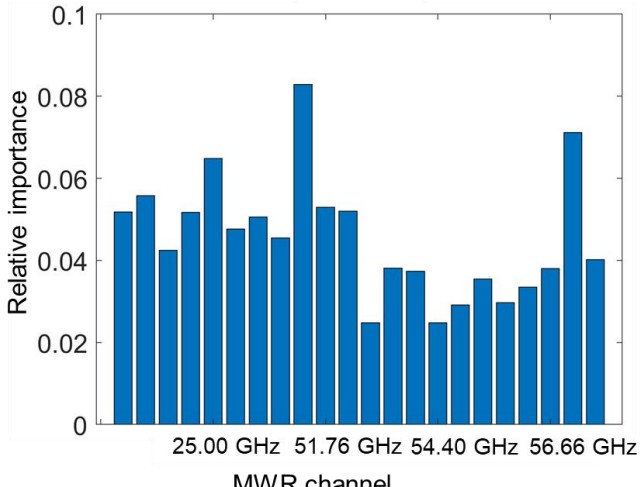


**Figure 2.** Relative importance of all BTs in different frequencies measured by MWR in the RFR model. The vertical axis represents relative importance (unitless), and the horizontal axis corresponds to different variable inputs (BTs in different frequencies measured by MWR in the RFR model). These channels include K band (22.23 GHz, 22.50 GHz, 23.03 GHz, 23.83 GHz, 25.00 GHz, 26.23 GHz, 28.00 GHz, 30.00 GHz) and V band (51.25 GHz, 51.76 GHz, 52.28 GHz, 52.80 GHz, 53.34 GHz, 53.85 GHz, 54.40 GHz, 54.94 GHz, 55.50 GHz, 56.02 GHz, 56.66 GHz, 57.29 GHz, 57.96 GHz, 58.80 GHz).








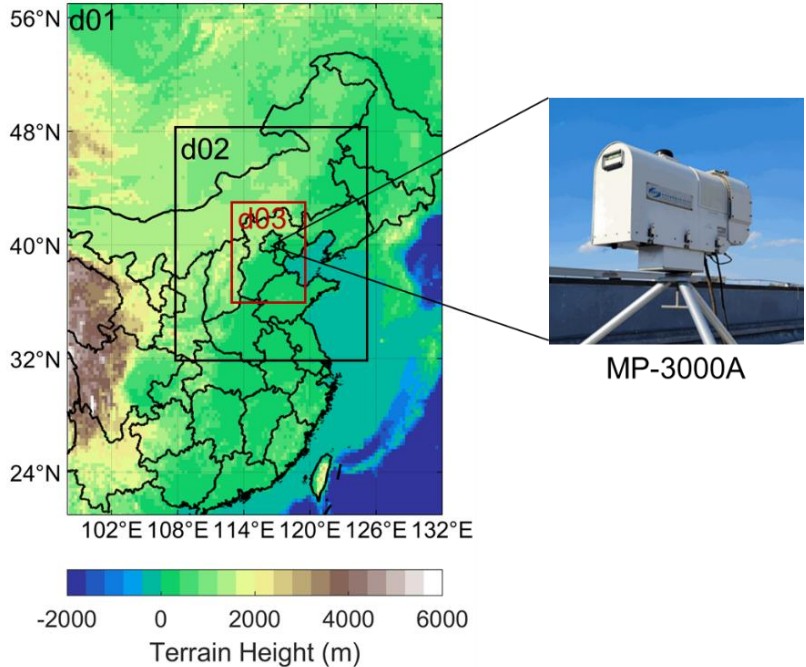


**Figure 3.** Simulation domains (left panel) of the WRF-Chem experiments. The MWR (right panel) used in this study is located in domain 3. This domain has a spatial resolution of 10 km. The MP-3000A MWR by Radiometrics is deployed at the Beijing Nanjiao Meteorological Observatory (39.80°N, 116.47°E) in China for brightness temperature (BT) measurements.

535




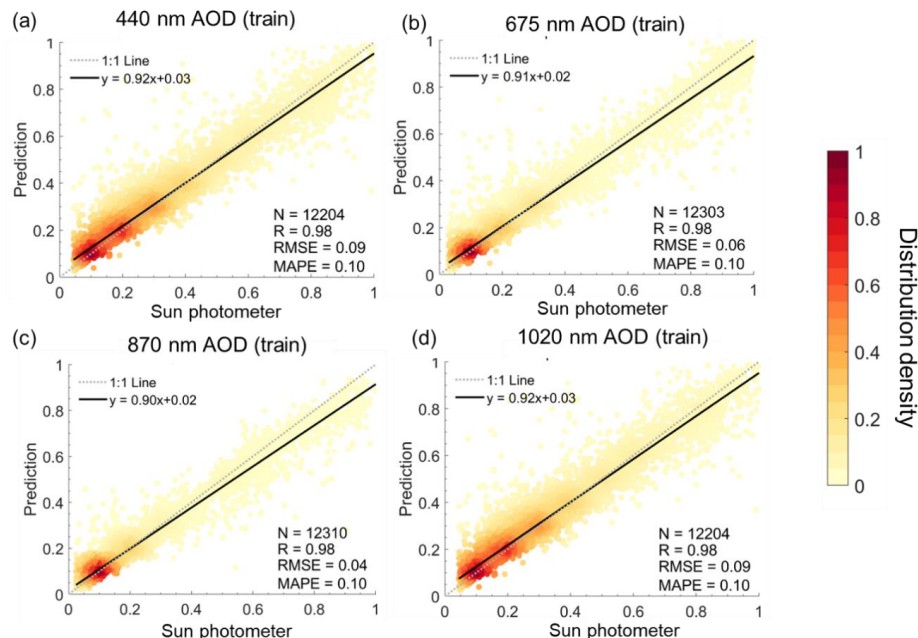

536

**Figure 4.** Density scatterplots of daytime AOD in the train set of MWR and sun photometer with (a) 440 nm, (b) 675 nm, (c) 870 nm, and (d) 1020 nm. The dashed dark gray line represents the 1:1 line, and the black solid line represents the linear regression line.

541

542





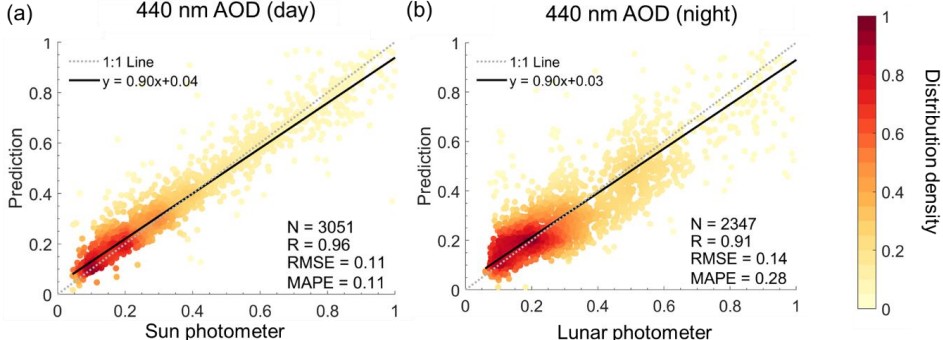

**Figure 5.** Density scatterplots of 440 nm AOD in the test set of MWR and the photometer with (a) daytime, and (b) nighttime. The dashed dark gray line represents the 1:1 line, and the black solid line represents the linear regression line. Note that the daytime corresponds to 6:00 am to 6:00 pm for the local time (UTC+8), and nighttime corresponds to the remaining time.



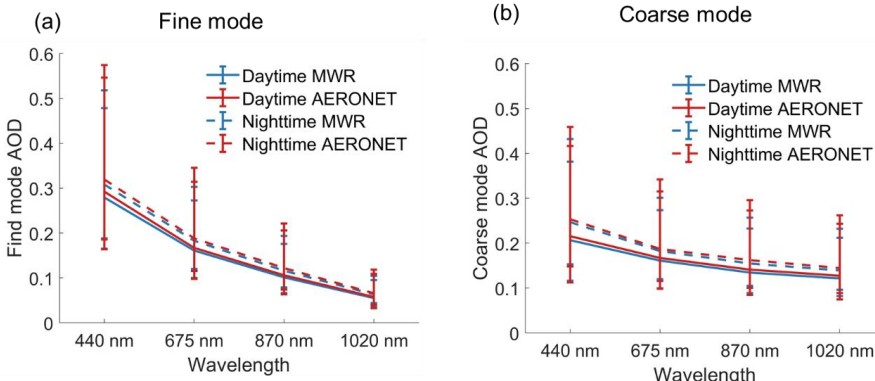

**Figure 6.** The relationship between wavelength and its corresponding AOD for MWR-based (blue lines) and the photometer (red lines) in the daytime (solid lines) and nighttime (dashed lines) for the (a) fine mode particles (440 nm to 870 nm Angstrom index > 1), and (b) coarse mode particles (440 nm to 870 nm Angstrom index < 1). The upper bound of the error bar is the 25th percentile, the middle is the median, and the lower bound is the 75th percentile.





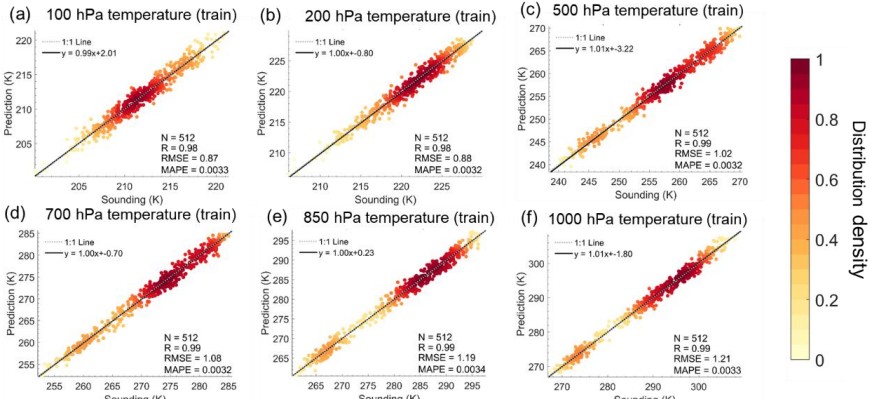

**Figure 7.** Density scatterplots of the vertical temperature profile in the train set of MWR and sounding data at (a) 100 hPa, (b) 200 hPa, (c) 500 hPa, (d) 700 hPa, (e) 850 hPa, and (f) 1000 hPa. The dashed dark gray line represents the 1:1 line, and the black solid line represents the linear regression line.





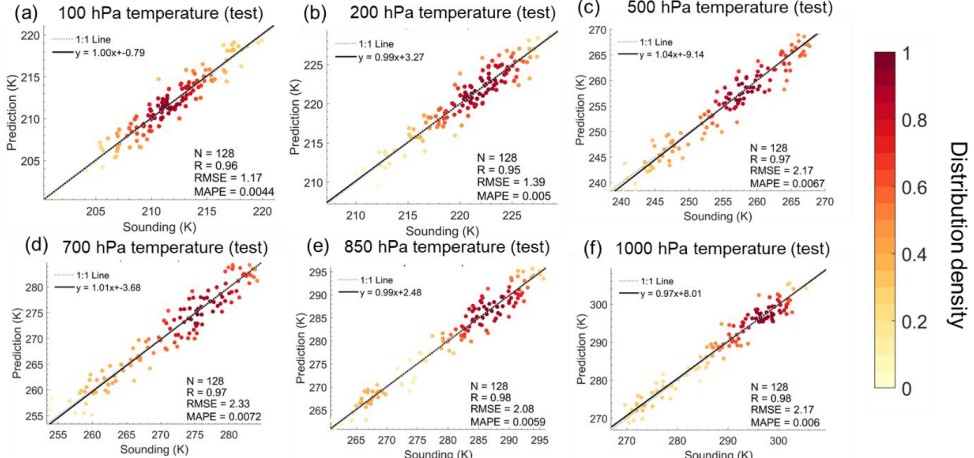

**Figure 8.** Density scatterplots of the vertical temperature profile in the test set of MWR and sounding data at (a) 100 hPa, (b) 200 hPa, (c) 500 hPa, (d) 700 hPa, (e) 850 hPa, and (f) 1000 hPa. The dashed dark gray line represents the 1:1 line, and the black solid line represents the linear regression line.





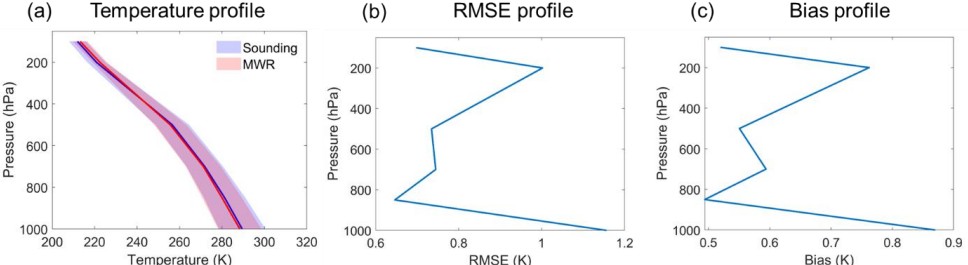

**Figure 9.** (a) Climatological mean vertical temperature profiles from sounding (the blue shading and line) and MWR (the red shading and line). (b) RMSE vertical profile calculated between sounding and MWR temperature, and (c) Similar to (b), but for the bias vertical profile.





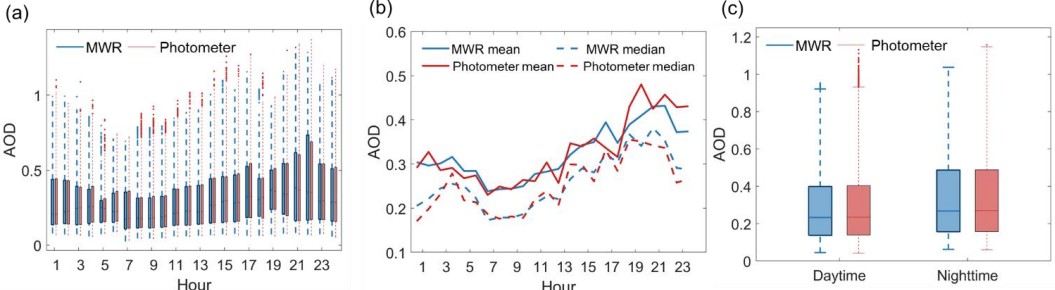

**Figure 10.** The diurnal cycle of MWR AOD and photometer AOD at 440nm. (a) The boxplot of hourly MWR AOD (red boxplots) and photometer AOD (blue boxplots). The small dots represent outliers greater than $q_{75} + 1.5(q_{75} - q_{25})$ or less than $q_{25} - 1.5(q_{75} - q_{25})$, where $q_{75}$ and $q_{25}$ correspond to 75th and 25th percentile. (b) The time series of mean AOD (solid lines) and median AOD (dashed lines) of MWR AOD (red lines) and photometer AOD (blue lines). (c) The boxplot of daytime and nighttime AOD. Blue boxes correspond to MWR data, and red boxes correspond to photometer data.



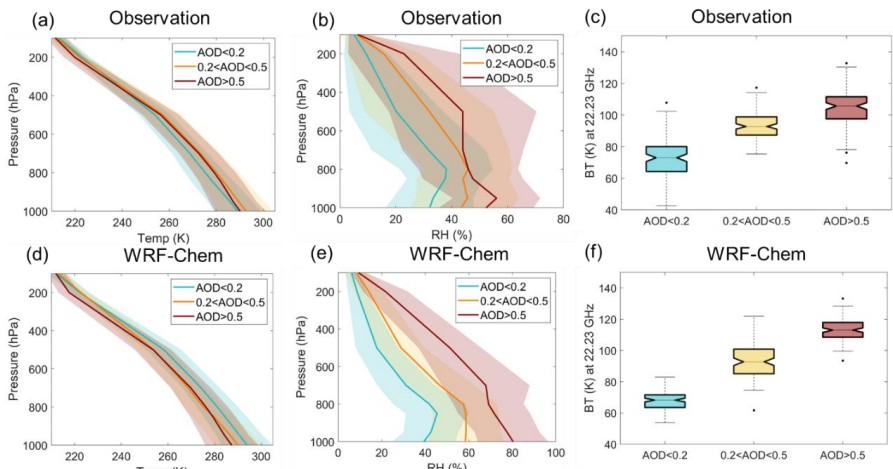

**Figure 11.** (a-b) The observational vertical profiles of temperature (Temp, unit: K) and relative humidity (RH, unit: %) under various AOD levels. The cyan, orange, and red solid lines correspond to low-level polluted scenarios (AOD<0.2), mid-level polluted scenarios (0.2<AOD<0.5), and high-level polluted scenarios (AOD>0.5). (c) Their corresponding brightness temperature (BT, unit: K) at 22.23 GHz calculated by MonoRTM. (d-f) Similar to a-c, but for the WRF-Chem simulation. The shadings represent the spread of samples with one standard deviation. All differences have passed the significance test of *p*-value≤0.01 by Student's *t*-test.





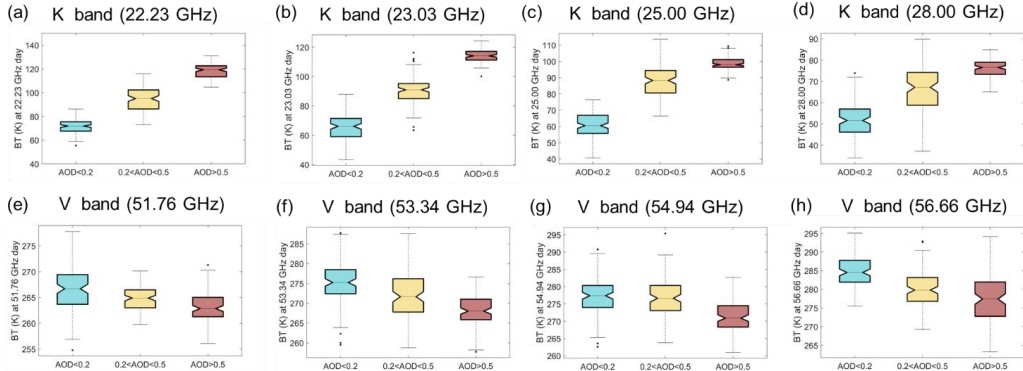

**Figure 12.** The boxplots of relationship between BT and AOD at 550 nm when fixing the surface temperature at 270-275 K from 0000 UTC 18 December 2016 to 0000 UTC 20 December 2016 in the WRF-Chem simulation. The frequencies of BT are (a) 22.23 GHz, (b) 23.03 GHz, (c) 25.00 GHz, (d) 28.00 GHz, (e) 51.76 GHz, (f) 53.34 GHz, (g) 54.94 GHz, and (h) 56.66 GHz during the daytime.





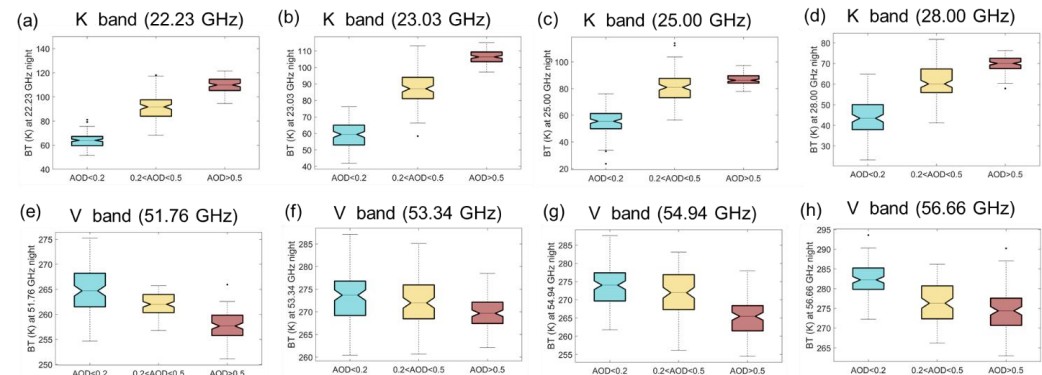

604

605 **Figure 13.** Similar to Figure 12, but for the nighttime.

606

607

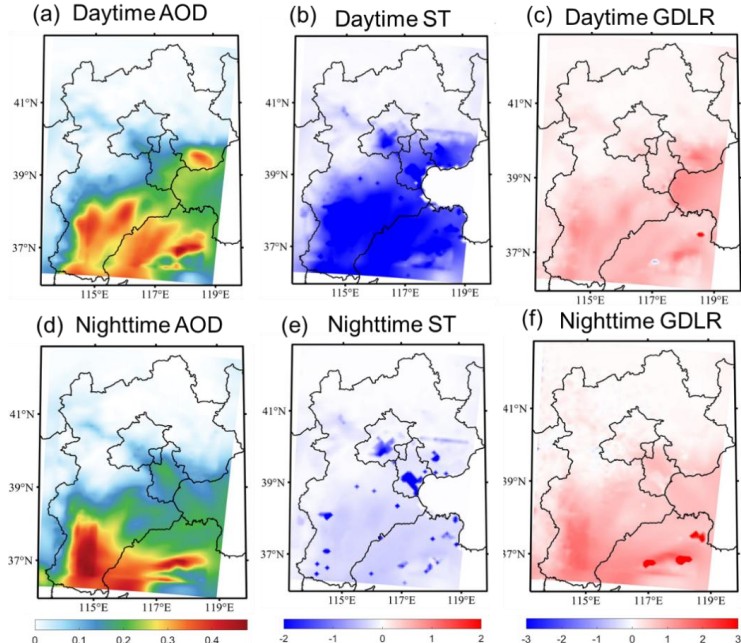

**Figure 14.** The difference of (a, d) aerosol optical depth (AOD), (b, e) surface temperature (ST), and (c, f) ground downward longwave radiation (GDLR) between EXP_AER and EXP_NOAER experiments (EXP_AER-EXP_NOAER) during the (a-c) daytime and (d-f) nighttime. The daytime corresponds to the period from 2200 UTC 18 December 2016 to 1000 UTC 19 December 2016. The nighttime corresponds to the period from 1000 UTC 19 December 2016 to 2200 UTC 19 December 2016.



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
