# Peer review of "Estimation of Nighttime Aerosol Optical Depths Using"

_EGUsphere, 2025_

## Author Comment (AC1)

**Response to the review of "Estimation of Nighttime Aerosol Optical Depths Using the Ground-based Microwave Radiometer" (EGUSPHERE-2025-1871).**

**Responses to Review 01 's comments**

Summary:

This study develops a novel microwave-based method for retrieving aerosol optical depth (AOD) using ground-based radiometer measurements at K- and V-bands. A random forest model trained with daytime sun-photometer data enables continuous day-night AOD monitoring, revealing higher nighttime values. Validation against lunar measurements, radiosonde data, and model simulations confirms the method's reliability. The approach overcomes traditional limitations of nighttime aerosol monitoring, providing new insights into diurnal AOD variations. This practical technique offers valuable applications for air quality assessment and climate studies, particularly for investigating nocturnal aerosol-cloud interactions and radiative effects. The operational simplicity and all-weather capability make it suitable for comprehensive environmental monitoring networks. While the study presents valuable findings, several aspects require further consideration prior to publication.

**Reply:** We are grateful to the reviewer for his/her insightful and very valuable suggestions, which have significantly contributed to the enhancement of our manuscript. We have carefully addressed the comments and made the necessary revisions, hoping that the updated version meets the journal's standards. Our responses to each of the comments and suggestions are as follows. The referee's original comments are shown in blue. Our replies are shown in black. The corresponding changes in the manuscript are shown in *Italic black.*

1. In Figure 2, the channel importance analysis in Figure 2 would be significantly improved by clearly labeling all channel names/frequencies. Additionally, please elaborate on the methodology used to calculate channel importance scores, as this is crucial for interpreting the variable selection process in your random forest model.

**Reply:** Thanks for your helpful suggestions. We have clearly labeled all frequencies in Figure 2 in the revised manuscript. We also cited it here for convenience (Figure R1).

[Figure]

*Figure R1. Relative importance of all BTs in different frequencies measured by MWR in the RFR model. The vertical axis represents relative importance (unitless), and the horizontal axis corresponds to different variable inputs (BTs in different frequencies measured by MWR in the RFR model). These channels include K band (22.23 GHz, 22.50 GHz, 23.03 GHz, 23.83 GHz, 25.00 GHz, 26.23 GHz, 28.00 GHz, 30.00 GHz) and V band (51.25 GHz, 51.76 GHz, 52.28 GHz, 52.80 GHz, 53.34 GHz, 53.85 GHz, 54.40 GHz, 54.94 GHz, 55.50 GHz, 56.02 GHz, 56.66 GHz, 57.29 GHz, 57.96 GHz, 58.80 GHz).*

Moreover, we also elaborated on the methodology used to calculate the channel importance scores for random forest regression models and cited it here (Lines 256-269).

*We first apply the relative importance feature selection technique, which is based on the Gini importance measure (Nembrini et al., 2018), to identify significant independent variables and build a generalized model. In the context of random forests,*

*the relative importance of each predictor variable (feature) is quantified by a numeric array of size 1-by-Nvars. The importance measure for each variable is defined as the increase in prediction error that results from permuting the values of that variable across the out-of-bag observations. This measure is calculated for each tree in the ensemble, then averaged across all trees. To standardize the importance scores, the average values are normalized by dividing them by the standard deviation computed over the entire ensemble. This process yields a normalized importance measure that provides a robust assessment of each feature's contribution to the model's predictive performance. The relative importance of each factor is presented in Figure 3. It is observed that BTs across various frequency bands carry similar levels of importance, suggesting that the BTs are almost equally important for retrieving AOD.*

2. The criteria for identifying clean sky cases in Figure 5 require more detailed explanation, particularly since this selection directly impacts your nighttime algorithm performance. Given the microwave sensor's limited sensitivity to cloud layers, please clarify: (a) your cloud screening methodology, and (b) how potential cloud contamination was addressed in the analysis. I think this algorithm developed in this study will be the operational algorithm during nighttime, the first step is to determine the clean sky cases.

Reply: Thanks for your suggestion and sorry for the confusion. The Level 2 sun photometer AOD products from AERONET are validated and represent clear-sky conditions. Therefore, collocating MWR data with these AERONET products inherently excludes cloudy conditions, ensuring that the most collocated data are under clear-sky scenarios. While AERONET data can be cloud-free in the direction of the sun, the MWR, which measures in the zenith direction, may still detect the presence of clouds. Therefore, we have conducted additional cloud screening following the method by the previous study to ensure the clear-sky conditions in the analysis (Zhang, 2024). We have revised the corresponding explanation in the methods section and cited it here (Lines 183-190).

*Notably, the Level 2 sun photometer AOD products from AERONET are already validated and represent clear-sky conditions. Therefore, the collocation of MWR data with these AERONET products inherently excludes cloudy conditions. While AERONET data can be cloud-free in the direction of the sun, the MWR, which measures in the zenith direction, may still detect the presence of clouds. Therefore, we have conducted additional cloud screening following the method by the previous study to ensure the clear-sky conditions in the analysis (Zhang, 2024).*

3. The physical interpretation in section 3.3 would benefit from incorporating established microwave scattering theory. Specifically, please discuss how your findings relate to known scattering and penetration characteristics of microwave channels for different aerosol particles, citing relevant literature (e.g., [reference 1], [reference 2]). This would strengthen the theoretical foundation of your approach.

Reply: Thanks for your insightful suggestions. The primary signals in the microwave spectrum that are attributable to aerosols are predominantly generated by the scattering and absorption properties of aerosols. Specifically, large aerosol particles play a significant role in this context. Moreover, the alterations in temperature and humidity profiles, which are closely linked to the radiative and hygroscopic effects of aerosols, also contribute to these signals. Indeed, previous studies have demonstrated that large aerosol particles, particularly dust aerosols, can significantly influence microwave radiation and BT (Ge et al., 2008; Hong et al., 2008), the primary mechanism by which MWR estimates AOD in this study might be through the detection of temperature and RH profile differences.

Therefore, we have added the detailed discussion in the section 3.3 about the microwave channels in the K bands to strengthen the theoretical foundation of our approach in the revised manuscript. We also cited it here for convenience (Lines 564-572).

*The increase of K band BT with AOD might be related to coherent changes of water*

*vapor and aerosols, either due to aerosol absorption of water or meteorological conditions that affect both water vapor and aerosols. When AOD is higher, RH is typically also higher, accompanied by more water vapor due to the hygroscopic growth effect of aerosols, as supported by previous analysis (Figure 11a & c). Since the K band includes the water vapor absorption line near 22.235 GHz, the BT in the K band is sensitive to water vapor, and thus the BT increases as AOD increases (Liu et al., 2014; Xie et al., 2013), further strengthening the theoretical foundation of the proposed approach.*

We also added the similar discussion in the section 3.3 about the microwave channels in the V bands and cited it here for convenience (Lines 581-586).

*The detailed physical interpretation as follows: due to the presence of the oxygen absorption band within the frequency range of the V band, it is highly sensitive to changes in atmospheric temperature (Van Leeuwen et al., 2001). Variations in AOD can influence the atmospheric temperature profile as shown by observation and simulation (Figure 11b &d). Consequently, in cases when AOD is high, the BT in the V band will decrease.*

We also added the detailed physical interpretation summary at the end of section 3.3 and cited it here (Lines 589-595).

*In conclusion, MWR has the potential to estimate AOD by identifying the differences in temperature and humidity profiles, as well as the direct scattering and absorption signals that arise from varying aerosol loadings. While previous studies have demonstrated that large aerosol particles, particularly dust aerosols, can significantly influence microwave radiation and BT (Ge et al., 2008; Hong et al., 2008), the primary mechanism by which MWR estimates AOD in this study might be through the detection of temperature and RH profile differences.*

4. Figure 10 current layout makes data interpretation challenging. I recommend reorganizing it using a 2×2 panel format to: (a) better separate day/night comparisons,

(b) improve visualization of temporal trends, and (c) allow direct visual comparison between different measurement types.

Reply: Thanks for your suggestions. We have modified the Figure 10 using a 3×1 panel format for better data interpretation. We also cited it here for convenience.

[Figure]

***Figure R2. The diurnal cycle of MWR AOD and photometer AOD at 500 nm.*** *(a) The boxplot of hourly MWR AOD (red boxplots) and photometer AOD (blue boxplots). The small dots represent outliers greater than $q_{75} + 1.5(q_{75} - q_{25})$ or less than $q_{25} - 1.5(q_{75} - q_{25})$, where $q_{75}$ and $q_{25}$ correspond to 75th and 25th percentile. (b) The time series of mean AOD (solid lines) and median AOD (dashed lines) of MWR AOD (red lines) and photometer AOD (blue lines). (c) The boxplot of daytime and nighttime AOD. Blue boxes correspond to MWR data, and red boxes correspond to photometer data.*

**References**

Ge, J., Huang, J., Weng, F., and Sun, W.: Effects of dust storms on microwave radiation based on satellite observation and model simulation over the Taklamakan desert, ATMOSPHERIC CHEMISTRY AND PHYSICS, 8, 4903-4909, 10.5194/acp-8-4903-2008, 2008.

Hong, G., Yang, P., Weng, F. Z., and Liu, Q. H.: Microwave scattering properties of sand particles: Application to the simulation of microwave radiances over sandstorms, J. Quant. Spectros. Radiat. Transfer, 109, 684-702, 10.1016/j.jqsrt.2007.08.018, 2008.

---

## Author Comment (AC2)

**Response to the review of "Estimation of Nighttime Aerosol Optical Depths Using the Ground-based Microwave Radiometer" (EGUSPHERE-2025-1871).**

**Responses to Review 02 's comments**

Summary:

This study presents the results of a new and innovative method based on microwave irradiances for retrieving Aerosol Optical Depth (AOD) at VIS/NIR wavelengths (440 nm, 675 nm, 870 nm, 1020 nm, and 550 nm). The method used the BT retrievals of a ground-based microwave radiometer (MWR) at K- and V-bands. The study has been led at the Beijing Nanjiao Meteorological Observatory in China, where the MWR instrument was operated.

The method (a machine learning based algorithm) is unfortunately not described. It is only informed, that a random forest model trained with daytime sun-photometer data has been used.

A validation of the results is presented comparing the AOD obtained with the new MWR method to the AOD measured with the photometer of the closest AERONET station (Beijing-CAMS) during day (solar photometry) and night (lunar photometry). The validation experiment has been run over at least 10 months from December 2019 to October 2020. The validation results show a good agreement between the new method and the photometry results.

The new method enables continuous day-night AOD monitoring in cloudless conditions. This is a gain for night measurements compared to the widespread lunar photometry technique that is restricted to the presence of the moon and the moon cycle.

As application, a study over diurnal (24 hours) AOD variations at the MWR site (Beijing Nanjiao Meteorological Observatory) is presented.

The text of the manuscript and the figures and caption is clear, the English is good. It is written in a simple and understandable way. The new method is welcome and innovative. The comparison/validation study shows convincing results. It is a manuscript of good quality showing a very good work.

**Reply:** We greatly appreciate the reviewer's insightful and very valuable suggestions, which have significantly contributed to the enhancement of our manuscript. We have carefully addressed the comments and made the necessary revisions, hoping that the updated version meets the journal's standards. Our responses to each of the comments and suggestions are as follows. The referee's original comments are shown in blue. Our replies are shown in black. The corresponding changes in the manuscript are shown in *Italic black.*

1.    Nevertheless, this paper has a relevance to be published because of the new and innovative method. The paper has a relevance to be published if this method is 1) described in details and precision and 2) if the validation of the method is presented. Unfortunately, even if the validation study is well presented (step 2), the first and most important step (presentation/description of the method) is missing. The method is not described but only presented as a black box: "machine learning based retrieval method focusing on the RFR method (Svetnik et al., 2003)". Before acceptance for publication, the section "Retrieval algorithm" has to be considerably extended with precise explanation with algorithm schemes and equations showing how the AOD in VIS/NIR wavelengths is extracted out of the Brightness Temperatures (microwave irradiances) retrieved by the MWR in the K-bands and V-bands. The paper must clearly explain on which atmospheric sciences physical processes between aerosol amount (AOD in VIS/NIR) and radiation in the microwave, the retrieval algorithm is based. Without this description, in my opinion, this article is not relevant for publication in AMT.

I suggest the authors of the manuscript to work again on it, develop an extended section about "Retrieval algorithm" given all the details about physical processes mentioned above and to submit again.

**Reply:** Thanks for your helpful suggestions. MWR has the potential to estimate AOD by identifying the differences in temperature and humidity profiles, as well as the direct scattering and absorption signals that arise from varying aerosol loadings. While previous studies have demonstrated that large aerosol particles, particularly dust aerosols, can significantly influence microwave radiation and BT(Ge et al., 2008; Hong et al., 2008), the primary mechanism by which MWR estimates AOD in this study might be through detecting the changes of temperature and RH profiles. More details concerning the physical basis of our retrieval algorithm can be referred to section 3.3, which provides a comprehensive physical interpretation.

We have also developed the extended section about "Retrieval algorithm". We also

cited it here for convenience.

*2.2 Retrieval Algorithm*

*Because the relationship between aerosol loading and microwave radiation is complicated and could be nonlinear, we use a machine learning based retrieval method focusing on the RFR method (Svetnik et al., 2003). The RFR model leverages the power of ensemble learning, integrating multiple decision trees to enhance prediction accuracy and robustness. Each decision tree within the ensemble is constructed using a random subset of the training data and a random selection of features, thereby reducing overfitting and improving generalization capabilities. Through this mechanism, the RFR model can effectively capture the complex interactions between aerosol properties and microwave radiation signals, providing a reliable and efficient approach for aerosol retrieval.*

*All variables are appropriately matched in both space and time. Specifically, AOD from sun photometer measurements and BTs from the MWR are matched within a 5-minute time window, while hourly temperature profiles from ERA-5 reanalysis datasets and BTs from the MWR are collocated within a 30-minute time window and a 15 km spatial radius.*

*We first apply the relative importance feature selection technique, which is based on the Gini importance measure (Nembrini et al., 2018), to identify significant independent variables and build a generalized model. In the context of random forests, the relative importance of each predictor variable (feature) is quantified by a numeric array of size 1-by-Nvars. The importance measure for each variable is defined as the increase in prediction error that results from permuting the values of that variable across the out-of-bag observations. This measure is calculated for each tree in the ensemble, then averaged across all trees. To standardize the importance scores, the average values are normalized by dividing them by the standard deviation computed over the entire ensemble. This process yields a normalized importance measure that*

provides a robust assessment of each feature's contribution to the model's predictive performance. The relative importance of each factor is presented in Figure 2. It is observed that BTs across various frequency bands carry similar levels of importance, suggesting that the BTs are almost equally important for retrieving AOD.

The retrieval algorithm is subsequently trained using eight selected K-band BTs and fourteen V-band BTs from the MP-3000A MWR as input variables. The target variables include AOD at 440 nm, 675 nm, 870 nm, and 1020 nm from the Beijing-CAMS AERONET site, as well as ERA-5 vertical temperature profiles at 100 hPa, 200 hPa, 500 hPa, 700 hPa, 850 hPa, and 1000 hPa. To ensure the representativeness of the sampling, we select the first 3/4 of the data in each month as the training set and the last 1/4 of the data as the testing set. Additionally, the algorithm is adapted to estimate nighttime AOD using nighttime BTs from microwave radiometry as inputs, which is then validated against nighttime AOD observations from lunar measurements in lunar photometer for the same period. Moreover, AOD, whether in the visible or microwave region, is associated with aerosol loading, which serves as the foundation for retrieving visible AOD using microwave observations. Since we primarily aim at retrieving AOD rather than aerosol type, we did not consider AOD at the other wavelengths when building the AOD retrieval model. The relationship between AOD at 550nm and that at the microwave band is enclosed in the random forest model. The model performance is assessed against photometer retrievals using metrics such as linear regression slope and intercept, correlation coefficient (R), root-mean-square error (RMSE), and mean absolute percentage error (MAPE).

The RFR model is built by varying the number of decision trees from 8 to 256. Through validation analysis, it is determined that the optimal number of trees is 128, based on the best performance during validation. After refining the algorithm through extensive training and testing, it is used to retrieve nighttime AOD from nighttime MWR BTs, with validation against collocated lunar AOD measurements from the lunar photometer. Moreover, before investigating the diurnal cycle of MWR derived

*AOD, we perform a quality control on the minute-resolution retrieval results that typically have a higher noise level. Specifically, for each specific minute, we extract the AOD for this minute from each day to form an AOD sequence. We then calculate the mean and standard deviation of this AOD sequence. Finally, we remove AOD that exceeds three times the standard deviation. Considering the suitable quantity of outliers procured by setting the threshold at three standard deviations and the prevalently utilized 3-sigma rule, we used three standard deviations as the threshold (Li et al., 2016a; Liu et al., 2024; Wang et al., 2012).*

2. In the introduction (Part 1): Increase your knowledges about the current reference technique regarding AOD measurements: The photometry using sun photometers during the day and lunar (widespread) or stellar (rare) photometers during the night. Cite relevant papers of these techniques. Be clear how you describe the restrictions of lunar photometry: Only the half of the nights because at least half moon is needed (moon cycles) / mostly only the half of each measurement night because the moon set/rise cycle is not anti-correlated to the sun set/rise cycle.

**Reply:** Thanks for your helpful suggestions. We have supplemented the introduction by adding more detail about the photometry. We also clarified the restrictions of the lunar photometry in the revised manuscript. We cited them here for convenience (Lines 64-76).

*Remote sensing of aerosol properties at night is a challenging task. Lunar photometer emerges during recent years as an effective and relative accurate nighttime AOD retrieval technique, and has been widely used within the AERONET since 2013 (Barreto et al., 2013; Barreto et al., 2016). However, this method is limited in its temporal coverage, providing data for only approximately half of each month. This limitation arises because the method requires a substantial amount of moon-reflected solar radiation, which is not consistently available due to the moon phase and the imperfect anti-correlation between the lunar and solar set/rise cycles (Barreto et al., 2017; Berkoff et al., 2011). Compared with the lunar photometer method, stellar*

*photometry, despite its rarity of use, provides nighttime AOD measurements by leveraging stellar irradiance, eliminating lunar phase corrections, with long-term datasets revealing diurnal aerosol dynamics (Pérez-Ramírez et al., 2011; Pérez-Ramírez et al., 2016; Pérez-Ramírez et al., 2008; Pérez-Ramírez et al., 2015).*

3. In the method presentation (Part 2): make a description of the sites where you make the measurements. For the three sites (MWR site, AERONET site and Radiosonde site), but of course most of all for the MWR site (Beijing Nanjiao Meteorological Observatory). Make a subparagraph explaining about the sites for each site (coordinates, position regarding the urban area of Beijing, expected aerosol/pollution, rural/urban/suburban site, generalities about the climate: cold/warm, wet/dry, cloudy/sunny in the different seasons), make a table summarizing the most important data (address, geocoordinates, distance to MWR site...) and show a map of Beijing Urban area with markers at the place of these three sites.

**Reply:** Thanks for your helpful suggestions. We have made the description of the sites where we make the measurement and a subparagraph explaining about the conditions of each site. Moreover, we have provided the table summarizing the data (Table S1 in the revised manuscript and Table R1 here) and showed the map with markers at the place of these three sites (Figure 2b in the revised manuscript and Figure R1b here).

We also cited them here for convenience (Lines 228-235 and Lines 152-160).

*In summary, our study primarily relies on in-situ measurements from three sites: the MWR site, the AERONET site, and the sounding site (see Table S1 and Figure 2b for details). These sites are located at the Beijing Nanjiao Meteorological Observatory, the Chinese Academy of Meteorological Sciences, and the Beijing Meteorological Station also in Beijing Nanjiao Meteorological Observatory, respectively. All three sites are situated within the urban or suburban areas of Beijing, with relatively close proximity to each other. The aerosol types expected at these sites include urban aerosols and mixed aerosols, with dust aerosols peaking during the boreal spring*

*season (Chen et al., 2016; Ou et al., 2017).*

*The study area is located at the northern edge of the North China Plain, featuring a temperate continental monsoon climate with four distinct seasons (Yu et al., 2009). Spring is occasionally influenced by dust episodes transported by northwesterly and westerly winds from the Kumutage and Taklimakan deserts in western China, or by northerly winds from the Mongolian deserts (Liu et al., 2022a). Summer is marked by relatively hot and humid conditions and accounts for approximately 74% of the annual precipitation. Autumn is mild and dry, with clear skies and cooling temperatures. Winter is cold and dry, with occasional snowfall and minimal precipitation (Feng et al., 2010; Hao et al., 2017).*

[Figure]

***Figure R1.*** *(a) Simulation domains of the WRF-Chem experiments. (b) Left panel: the locations of observation sites in this study. The black circle indicates the MWR, the dark grey square indicates the photometer, and the light triangle indicates the sounding. Right panel: the MWR used in this study is located in domain 3. This domain has a spatial resolution of 10 km. The MP-3000A MWR by Radiometrics is deployed at the Beijing Nanjiao Meteorological Observatory (39.80°N, 116.47°E) in China for brightness temperature (BT) measurements.*

***Table R1.*** *The detailed locations of the measurement sites*

|  | MWR site | AERONET site | Sounding site |
|---|---|---|---|
| Geocoordinates | 39.80 °N, 116.47 °E | 39.95 °N, 116.32°E | 39.80°N, 116.48°E |
| Address | Beijing Nanjiao Meteorological Observatory | Chinese Academy of Meteorological Sciences | Beijing Meteorological Station |
| Location type | Suburban | Urban | Suburban |
| Distance to MWR site (km) | 0 | 20.77 | 1.11 |

4. In the validation study (Part 3): You must know that the reference wavelengths for the photometry is 500 nm (for satellite remote sensing, the reference wavelength is 550 nm). When you make a validation study against photometers, you must show results for this channel.

**Reply:** Thanks for your helpful suggestions and you are right. We have shown the results at 500 nm in the revised version and mainly discuss the model's performance at this wavelength. We also cited the updated Figures here for convenience.

[Figure]

***Figure R2.*** *Density scatterplots of daytime AOD in the train set of MWR and sun photometer with (a) 440 nm, (b) 500nm, (c) 675 nm, (d) 870 nm, and (e) 1020 nm. The dashed dark gray line represents the 1:1 line, and the black solid line represents the linear regression line.*

[Figure]

***Figure R3.*** *Density scatterplots of 500 nm AOD in the test set of MWR and the photometer with (a) daytime, and (b) nighttime. The dashed dark gray line represents the 1:1 line, and the black solid line represents the linear regression line. Note that the daytime corresponds to 6:00 am to 6:00 pm for the local time (UTC+8), and nighttime corresponds to the remaining time.*

[Figure]

***Figure R4.*** *The relationship between wavelength and its corresponding AOD for MWR-based (blue lines) and the photometer (red lines) in the daytime (solid lines) and nighttime (dashed lines) for the (a) fine mode particles (440 nm to 870 nm Angstrom index > 1), and (b) coarse mode particles (440 nm to 870 nm Angstrom index < 1). The upper bound of the error bar is the 25th percentile, the middle is the median, and the lower bound is the 75th percentile.*

[Figure]

***Figure R5.*** *The diurnal cycle of MWR AOD and photometer AOD at 500 nm. (a) The boxplot of hourly MWR AOD (red boxplots) and photometer AOD (blue boxplots). The small dots represent outliers greater than* $q_{75} + 1.5(q_{75} - q_{25})$ *or less than* $q_{25} - 1.5(q_{75} - q_{25})$, *where* $q_{75}$ *and* $q_{25}$ *correspond to 75th and 25th percentile. (b) The time series of mean AOD (solid lines) and median AOD (dashed lines) of MWR AOD (red lines) and photometer AOD (blue lines). (c) The boxplot of daytime and nighttime AOD. Blue boxes correspond to MWR data, and red boxes correspond to photometer data.*

5.  For your application study, confrontate your results to the expected results regarding AOD load, and compare it (at least qualitatively) to former studies

**Reply:** Thanks for your valuable suggestions. We have compared our results with the expected AOD observation (photometer) and compared it with former studies. We also added this here for reference (Lines 450-452 and Lines 480-502).

*Although the MWR-based AOD tends to underestimate extreme values relative to the photometer AOD, the MWR measurements exhibits strong agreement with the photometer AOD.*

*We have further divided the results into four seasons and validated that the diurnal cycle of AOD is consistent across all seasons, with the most pronounced diurnal difference occurring in summer (Figure S1). It is noted that the lunar AOD is not available for JJA, which further underscores the supplementary role of MWR-predicted AOD in complementing lunar AOD measurements. The seasonal variation of AOD diurnal cycle agrees with previous studies derived from downscaling reanalysis datasets (Wang et al., 2025). The more significant diurnal difference in AOD during summer can be attributed to two primary factors. Firstly, the intense solar radiation and high temperatures prevalent in summer significantly promote the formation of aerosol particles through the process of gas-to-particle conversion (Chen et al., 2023a). Secondly, the high humidity levels in summer facilitate aerosol hygroscopic growth, which enhances aerosol extinction (Chen et al.,*

*2023b; Lv et al., 2017). AOD at the other wavelengths (440 nm, 675 nm, 870 nm, and 1020 nm) exhibit similar diurnal patterns with peaks at about 20:00-22:00 (not shown here) and higher nighttime AOD in general (Figure 6).*

*The increase in nighttime AOD compared to daytime can be attributed to various factors, including a shallower mixed layer due to reduced horizontal mixing and transport, a decrease in atmospheric environmental capacity, higher relative humidity, enhanced aerosol hygroscopic growth, or intensified pollution emissions (Brock et al., 2016). Similar observations of elevated nighttime particle matter concentration have been reported in previous studies (Perrone et al., 2022; Su et al., 2023). However, research on nighttime aerosol properties is limited, warranting further analysis to fully understand these discrepancies, which exceeds the scope of this study.*

GENERALITIES

6. Please add a table with algorithms and parameters, this helps the understanding of the reader

Reply: Thanks for your insightful suggestions. We have added the table with algorithms and parameters (Table S2 in the revised manuscript), this helps the understanding of the reader. We also cited this table here for convenience (Table R2).

*Table R2. The parameters for the random forest regression model*

| Parameters | Interpretation | Value |
|---|---|---|
| Predictor variable | MWR BT observation at 22 K and V bands | BT |
| Predicted variable | AOD at five wavelengths and temperature profiles | AOD and temperature |
| Number of | The number of decision trees in the ensemble. | 128 |

trees

| | | |
|---|---|---|
| Minimum leaf size | Minimum number of observations per tree leaf. | 5 |
| In bag fraction | Fraction of observations that are randomly selected with replacement for each bootstrap replica. | 1.0 |
| Max features | Number of predictor or feature variables to select at random for each decision split. | 5 |
| Prune | The Prune property is true if decision trees are pruned and false if they are not. | False |

ABSTRACT

7.  L9-11" However, existing remote sensing methods mostly rely on the shortwave spectrum, which does not allow measurements at nighttime"

-> Wrong assumption. Today is lunar photometry very widespread (see hundreds of AERONET stations... much more than MWR stations!). MWR measurements should be presented as a complement

**Reply:** Thank you for your insightful and valuable feedback. We apologize for the inaccurate assumption in our initial statement. You are correct that lunar photometry is now widely utilized, as evidenced by the extensive network of AERONET stations (which significantly outnumber MWR stations). We acknowledge that MWR measurements should be presented as a complementary technique. But the lunar photometer is restricted because the method requires a substantial amount of moon-reflected solar radiation, which is not consistently available due to the moon phase and the imperfect anti-correlation between the lunar and solar set/rise cycles (Barreto et al., 2017; Berkoff et al., 2011).

Accordingly, we have revised the relevant expression in our manuscript and have cited it here for convenience (Lines 8-13).

*Aerosol optical depth (AOD) is a crucial parameter for understanding the impact of aerosols on Earth's atmosphere and air quality. Nevertheless, most existing remote sensing techniques rely on the shortwave spectrum, precluding nighttime measurements. While lunar and stellar photometry can measure nighttime AOD, their data availability is limited due to the scarce moonlight for lunar photometry and the rarity of application for stellar photometry.*

8. Precise at which site you make this comparisons/measurements ---> Beijing Nanjiao Meteorological Observatory in China?

**Reply:** Thanks for your suggestion, we have clarified the measurement and comparison locations precisely and cited it here (Lines 13-15 and Lines 19-21).

*In this study, we made a first attempt to retrieve AOD from ground-based microwave radiometer (MWR) measurements in Beijing Nanjiao Meteorological Observatory in China.*

*The algorithm demonstrates satisfactory performance, with strong agreements with lunar AOD retrievals from the lunar photometer (R=0.91 and RMSE=0.14).*

9. L17-18: "The algorithm demonstrates satisfactory performance, with strong agreements with lunar AOD retrievals." -> Give some statistical (RMSE or MBE or correlation ...) values to quantify the assumption "strong agreements".

**Reply:** Thanks for your suggestion. We have changed the wording and added statistical values to quantify this assumption in the revised manuscript and cited them here (Lines 19-21).

*The algorithm demonstrates satisfactory performance, with reasonable agreements with lunar AOD retrievals from the lunar photometer (R=0.91 and RMSE=0.14).*

10. L18-19: The results also reveal a distinct day-night cycle of AOD, with nighttime AOD typically higher than its daytime value" -> In general or for the station considered? Which station/site?

**Reply:** Thanks for your question and sorry for the confusion caused. This statement is based on the station considered in our study, that is the AOD from the Beijing-CAMS AERONET station and AOD estimated based on brightness temperature measurement of Beijing Nanjiao Meteorological Observatory in China. Moreover, our previous study also uncovered the nighttime AOD increase by retrieving nighttime AOD by utilizing longwave measurements in the atmospheric window region from the Atmospheric InfraRed Sounder (AIRS) instrument, further corroborated by surface and space lidar measurements (Liu et al., 2024). Therefore, this phenomenon might be general but still need to be validated using more extensive nighttime AOD measurements.

Therefore, we have clarified this statement in the abstract and cited it here (Lines 21-24).

*The results also reveal a distinct day-night cycle of AOD, with nighttime AOD typically higher than its daytime value for the Beijing-CAMS Aerosol Robotic Network (AERONET) site and AOD estimated based on MWR measurements.*

1. Introduction

11. L31-31: Please regarding the challenge of estimating aerosol radiative forcing cite the error bars of the last IPCC reports. These are common worldwide references

**Reply:** Thanks for your suggestions. We have cited this IPCC reports in this statement and cited it here (Lines 34-36).

*However, accurately assessing their role in radiative forcing is a major challenge (Fan et al., 2016; Ghan et al., 2016; IPCC, 2021; Seinfeld et al., 2016).*

12. L34: "optically" -> from its direct radiative impact

**Reply:** Thanks for your suggestions. We have revised as suggested.

13. L37-34: You first need to explain that AOD is obtained inverting the Beer-Bougher-Lambert equation of spectral direct normal irradiance (DNI) attenuation, usually using a spectrometer or a spectroradiometer making a direct sun irradiance observation most monochromatically on a spectral channel as possible with a filter (photometer) or a narrow band (spectroradiometer).-> This is the reference equation, method and instrumentation. Then you can describe other methods.

**Reply:** Thanks for your suggestions and you are right. Thus, we firstly explained as suggested and cited it here (Lines 41-46).

*The AOD is firstly measured through the inversion of the Beer-Bouguer-Lambert law, which describes the attenuation of spectral direct normal irradiance (DNI) (Gueymard, 2012). This process typically involves the use of a spectrometer or spectroradiometer to measure direct solar irradiance as monochromatically as possible on a specific spectral channel (Gueymard, 2012). This can be achieved using either a filter-based photometer or a narrow-band spectroradiometer.*

14. L42-43: "thus only daytime AOD can be obtained" -> This is wrong: There is stellar photometry with star photometers since the 90ies (mention it and cite references) and since 2013 with Cimel CE318T photometer (AERONET network instrument), lunar photometry (cite papers of Barreto et al., also 2013). You can point out, that stellar photometry is not well widespread (only a few stations worldwide due to bulky facilities and complicate operational process) and that lunar photometry has restriction (only the half of the nights because of the cycles, and mostly not complete nights because sun set and moon rise are not timely corresponding), but you cannot mention that there is no AOD measurement during the night possible with the standard well proved reference method that is photometry.

**Reply:** Thanks for your insightful and helpful suggestions. We have revised these statements and cited them here for convenience (Lines 64-76).

*Remote sensing of aerosol properties at night is a challenging task. Lunar photometer*

*emerges during recent years as an effective and relative accurate nighttime AOD retrieval technique, and has been widely used within the AERONET since 2013 (Barreto et al., 2013; Barreto et al., 2016). However, this method is limited in its temporal coverage, providing data for only approximately half of each month. This limitation arises because the method requires a substantial amount of moon-reflected solar radiation, which is not consistently available due to the imperfect anti-correlation between the lunar and solar set/rise cycles (Barreto et al., 2017; Berkoff et al., 2011). Compared with the lunar photometer method, stellar photometry, despite its rarity of use, provides reliable nighttime AOD measurements by leveraging stellar irradiance with long-term datasets revealing diurnal aerosol dynamics from 1990s (Pérez-Ramírez et al., 2011; Pérez-Ramírez et al., 2016; Pérez-Ramírez et al., 2008; Pérez-Ramírez et al., 2015).*

15. L55: OK now you mention lunar photometry. Good but 1) you forgot to mention solar photometry as first and reference method at the beginning of the introduction, and 2) you say now that we can monitor AOD during the night with photometers what is refuting your sentence of L42-43. Please restructure introduction: 1) Explain AOD as you did 2) Explain Beer-Lambert-Bougher equation and talk about photometry: solar photometry (since the 80ies) sun photometer has to be mention and the prior papers have to been cited. 3) Talk about photometry during the night (lunar and stellar) 4) explain the weaknesses/restrictions of lunar and stellar photometry that justify the use of new techniques for instance MWR that you will develop in this paper.

**Reply:** Thanks for your insightful and helpful suggestions. We have reconstructed the introduction section according to your suggestion. Please refer to the introduction section in the revised manuscript.

16. L63-65: Cite also older publications regarding stellar photometry:

-> HERBER, A., THMASON, L.W., GERNANDT, H., LEITERER, U., NAGEL, D., SCHULZ, K.H., KAPTUR, J., ALBRECHT, T. and NOTHOLT, T., 2002, Continuous day and night aerosol optical depth observations in the Artic between 1991 and 1999.

Journal of Geophysical Research, 107, p. 4097

-> LEITERER, U., NAEBERT, A., NAEBERT, T. and ALEKSEEVA, G., 1995, A new star photometer developed for spectral aerosol optical thickness measurements in Lindenberg. Contributions to Atmospheric Physics, 68, pp. 133–141.

**Reply:** Thanks for your suggestions and we have cited these publications.

17. L62 "eliminating lunar phase corrections" -> This sentence is confusing, we understand that you correct the lunar radiation what is not the case. Better withdraw it.

**Reply:** Thanks for your suggestions. We have deleted this sentence.

18. L72: Please precise that VIIRS is a satellite-based instrument (before you were describing ground base instruments)

**Reply:** Thanks for your suggestions. We have clarified that VIIRS is a satellite-based instrument and cited it here (Lines 84-87).

*For example, Zhang et al. examined the effectiveness of retrieving nighttime AOD over urban areas by utilizing city lights observed through the satellite-based instrument VIIRS (Visible Infrared Imaging Radiometer Suite) Day-Night Band (DNB) (Zhang et al., 2019).*

19. L84-85 if you compare day and night AOD, please mention Grassl et al. 2024 and her study of the Arctic: Graßl, S., Ritter, C., Wilsch, J., Herrmann, R., Doppler, L., & Román, R. (2024). From Polar Day to Polar Night: A Comprehensive Sun and Star Photometer Study of Trends in Arctic Aerosol Properties in Ny-Ålesund, Svalbard. Remote Sensing, 16(19), 3725.

**Reply:** Thanks for your suggestions. We have mentioned it in the revised manuscript and cited it here (Lines 101-103).

*Grassl et al. (2024) also presented a homogenized dataset derived from a sun and star photometer operated in the European Arctic over a 20-year period.*

2. Data and Methods

**Reply:** Thanks for your suggestions. We have given more information about this site and cited them here (Lines 161-163, Table S1 in the revised manuscript and Table R3 here).

*In this study, we utilized BT data collected from the MP-3000A MWR, which was stationed at the Beijing Nanjiao Meteorological Observatory located in China (39.80°N, 116.47°E, http://bj.cma.gov.cn/) (Ding et al., 2010; Lei et al., 2011; Zhou et al., 2024).*

***Table R3.*** *The detailed locations of the measurement sites*

| | MWR site | AERONET site | Sounding site |
|---|---|---|---|
| Geocoordinates | 39.80°N,116.47°E | 39.95°N,116.32°E | 39.80°N,116.48°E |
| Address | Beijing Nanjiao Meteorological Observatory | Chinese Academy of Meteorological Sciences | Beijing Meteorological Station |
| Location type | Suburban | Urban | Suburban |
| Distance to MWR site (km) | 0 | 20.77 | 1.11 |

 Explain also in which area the site of the measurements are (urban, suburban, rural, flat lands, mountains, forests, green fields, ...)

**Reply:** Thank you for your suggestions. We have detailed the site area of the measurement in Table S1 of the revised manuscript (Table R2 here).

22. L149 We use the data ranging from December 2019 to October 2020 -> Why so old data?

**Reply:** Thank you for your suggestions. We apologize for the limitation in the time range of the data acquisition. Due to constraints in the data distribution of China Meteorological Administration to which the Nanjiao cite belongs to, we were only able to obtain data for the specified period. We also aim to extend the time range of our analysis in the future study.

We have provided a detailed explanation of this limitation in the revised manuscript and have cited it here for clarity (Lines 172-175).

*We use the data ranging from December 2019 to October 2020 with a temporal resolution of one minute due to limitations of data distribution policy. We also aim to extend the temporal range of our analysis in the future study.*

23. L151-156: Explain how much data (in proportions) you needed to flag out because of instrumental faults and calibration problems and environmental factors.

Reply: We have carefully reviewed the data and found that 4.36% of the brightness temperature (BT) data were flagged out due to instrumental faults, calibration issues, and environmental factors. We have included a detailed explanation of these data exclusions in the revised manuscript and have cited it here for reference (Lines 181-183).

*Ultimately, nearly 4.36% of BT data were excluded from the study due to a combination of instrumental faults, calibration problems, and environmental factors.*

24. L157-159: "Notably, because the collocation between MWR and Level 2 sun photometer AOD products from the AERONET is already clear-sky data, there is no

need to perform cloud screening on the MWR data." -> This is most of the time correct but not always! You can be cloud free in the direction of the moon or the sun for AERONET data but if you look to zenith like the MWR does, there can be some clouds

**Reply:** Thanks for your kind reminds and you are right. We have conducted additional cloud screening following the method by the previous study (Zhang et al., 2024). This method considers the relative humidity as the criteria to ensure clear-sky conditions. Therefore, we have modified the statement and corresponding explanation and cited it here (Lines 186-190).

*While AERONET data can be cloud-free in the direction of the sun, the MWR, which measures in the zenith direction, may still detect the presence of clouds. Therefore, we further conducted an additional cloud screening following the method by the previous study to ensure the clear-sky conditions in the analysis (Zhang, 2024).*

25. L160: "Beijing-CAMS AERONET" -> Please precise the coordinates of the site, the distance to the site with the MWR instrument and the environment of the station

**Reply:** Thank you for your insightful suggestions. In response, we have meticulously refined the details in the revised manuscript to include the precise coordinates of the site, the exact distance between the site and the location of the MWR instrument, as well as a comprehensive description of the environmental characteristics of the station. These updated details have been appropriately cited within the text for reference (Lines 191-194).

*AOD retrieved using the solar and lunar methods at the Beijing-CAMS AERONET site (39.95°N,116.32°E, located in the Chinese Academy of Meteorological Sciences, see Table S1), which is the closest site to the MWR location (20.77 km), is used as training and validation data in the retrieval algorithm.*

26. -> I guess it is at 26 km distance (regarding AERONET coordinates and coordinates), this is far away in an urban area for a comparison. Why didn't you set the MWR directly at Beijing-CAMS station for the comparative study? Or at one of

the three other AERONET station of Beijing? At least you should make a spatial heterogeneity study showing how AOD differs from one Beijing AERONET site to the other during the 10 months comparison.

**Reply:** Thank you for your valuable suggestions. The distance between the two sites is 20.77 km. Considering the vast urban area of Beijing, which spans approximately 160 km both east-west and north-south, this distance is relatively short. Moreover, we also investigate the detailed AOD distribution in this area with finer resolution (0.05° × 0.05°) products derived from the MODIS satellite observation (Figure R6). It is noticed that the pollution levels and AOD values at the various sites within the urban area of Beijing are found to be relatively consistent. This finding underscores the uniformity of AOD spatial distribution throughout the urban expanse of Beijing, indicating that despite the overall gradient trend, the urban area exhibits a relatively homogeneous AOD pattern.

For a detailed visualization of the spatial distribution, please refer to Figure R7b. We specifically chose this AERONET station over others because it is the only one that provides consistent Level-2 data from 2019 to 2020, ensuring a reliable and uninterrupted dataset for our analysis.

[Figure]

**Figure R6.** The annual mean MODIS AOD distribution over Beijing from the

MCD19A2CMG product. The black circle indicates the MWR, the dark grey square indicates the photometer, and the light triangle indicates the sounding.

We have also added the explanation in the revised manuscript and cited it here (Lines 197-203).

*It is noteworthy that the distance between the Beijing-CAMS AERONET site and MWR site is 20.77 km. Considering the vast urban area of Beijing, which spans approximately 160 km both east-west and north-south, this distance is relatively short. We specifically chose this AERONET station other than others because it is the only one that provides consistent Version 3 Level 1.5 lunar AOD products from 2019 to 2020, ensuring a consistent dataset with daytime AOD for our analysis.*

[Figure]

***Figure R7.*** *(a) Simulation domains of the WRF-Chem experiments. (b) Left panel: the locations of observation sites in this study. The black circle indicates the MWR, the dark grey square indicates the photometer, and the light triangle indicates the sounding. Right panel: the MWR used in this study is located in domain 3. This domain has a spatial resolution of 10 km. The MP-3000A MWR by Radiometrics is deployed at the Beijing Nanjiao Meteorological Observatory (39.80°N, 116.47°E) in China for brightness temperature (BT) measurements.*

27. L163: Please give the coordinates and the distance to AERONET station CAMS Beijing and also to the station of the MWR instrument.

**Reply:** Thank you for your valuable suggestions. We have given the coordinates and the distance in the revised manuscript and cited it here (Lines 197-200).

*It is noteworthy that the distance between the Beijing-CAMS AERONET site and MWR site is 20.77 km. Considering the vast urban area of Beijing, which spans approximately 160 km both east-west and north-south, this distance is relatively short.*

28. L170: Why do you use ECMWF products, isn't it better directly train the MWR algorithm with the Radiosonde data.

**Reply:** Thank you for your valuable suggestions. We opted for the ECMWF products primarily due to their finer temporal resolution (hourly), which is contains much more detailed information than the sounding data (twice daily). This higher temporal resolution translates to a greater number of training samples for the Random Forest Regression (RFR) model, thereby enhancing its ability to capture the temporal variability and improve prediction accuracy of the retrieved variables. To further evaluate the accuracy of the RFR model in predicting vertical temperature profiles, we utilized collocated sounding data obtained from the Beijing Meteorological Station (station ID: 54511) during the corresponding time frame. This additional validation step ensures a comprehensive assessment of the model's performance against independent, high-quality measurements.

We further added the detailed explanation in the revised manuscript and cited it here (Lines 208-212).

*We chose the ECMWF products mainly because of their hourly temporal resolution, which provides more training samples for the RFR model than the twice-daily sounding data. This enhances the model's ability to capture temporal variability and improve prediction accuracy of the predicted variables.*

29. + Give references for ECMWF ERA-5

**Reply:** Thank you for your valuable suggestions. We have added the reference and cited it here (Lines 206-208).

*This is achieved by using temperatures at different pressure levels obtained from the European Center for Medium-Range Weather Forecasts (ECMWF) Reanalysis version 5 (ERA-5) as the target for our training (Hersbach, 2023).*

30. Why don't you mention this profile measurements in the Figure 1 or in its caption? Don't you use them for the training?

**Reply:** Thank you for your valuable suggestions and sorry for the ignoration. Indeed, we used the temperature profile measurements for the training, thus we added them in the Figure 1 and cited it here for convenience (Figure R8).

[Figure]

***Figure R8.*** *The flowchart for clear sky nighttime AOD and vertical temperature profiles retrieval algorithm.*

31. L180: Give references for MonoRTM

**Reply:** Thank you for your valuable suggestions, and we have given the reference for MonoRTM as suggested.

32. L188-189: "15 km spatial radius"? For ERA-5? in the way it is formulated: Last method/instrument mention is MWR, we understand that MWR has a radius of 15 km. It can only be the model. therefore please reformulate it clearly. By the way, since the AERONET station and the MWR stations are more than 25 km away, there are not in the same model grid cell if this one has a radius of 15 km and centered on one station = you obtain different model results for AERONET station and MWR station. How do you deal with this?

**Reply**: Thank you for your question and we apologize for the confusion. In fact, hourly temperature profiles from the ERA-5 reanalysis datasets are collocated with MWR BTs within a 30-minute time window and a 15 km spatial radius. The acquisition of temperature profiles relies solely on the ERA-5 reanalysis data and does not require data from the AERONET station. Consequently, the 15 km spatial radius refers to the distance between the ERA-5 grid point and the MWR site location. We thus clarified this statement in the revised manuscript and cited it here for convenience (Lines 247-255).

*All variables are rigorously matched in both temporal and spatial dimensions to ensure consistency and accuracy. Specifically, AOD data derived from sun photometer measurements are temporally matched with BTs from the MWR within a 5-minute time window. Meanwhile, hourly temperature profiles from the ERA-5 reanalysis datasets are collocated with MWR BTs within a 30-minute time window and a 15 km spatial radius. It should be noted that the acquisition of temperature profiles relies solely on the ERA-5 reanalysis data and does not require data from the AERONET station, and that the 15 km spatial radius only refers to the distance between the ERA-5 grid point and the MWR site location.*

33. L197-198: Why don't you consider the most important (because reference) wavelength of the AOD: 500 nm?

**Reply:** Thanks for your helpful suggestions and you are right. We have shown the

results at 500 nm in the revised version and mainly discuss the model's performance at this wavelength. We also cited the revised statement here for convenience (Lines 270-274).

*The retrieval algorithm is subsequently trained using eight selected K-band BTs and fourteen V-band BTs from the MP-3000A MWR as input variables. The target variables include AOD at 440 nm, 500 nm, 675 nm, 870 nm, and 1020 nm from the Beijing-CAMS AERONET site, as well as ERA-5 vertical temperature profiles at 100 hPa, 200 hPa, 500 hPa, 700 hPa, 850 hPa, and 1000 hPa.*

34. L196-212 + Figure 1: It is not clear: IS AERONET AOD used for the training or only for verification? Is it the same for day and night?

**Reply**: Thank you for your question. The AERONET AOD is utilized for both training and verification purposes. Specifically, daytime AERONET AOD data are employed to train and test the model. Once the model is trained, nighttime MWR BT measurements are input into the model to generate nighttime AOD estimates as the output. These model-derived nighttime AOD values are subsequently compared with the corresponding nighttime AERONET AOD measurements for validation. We also added the explanation part at the beginning of the section 3.1 to make it clearer (Lines 376-382).

*The AERONET AOD data are used for training and validating the model. Specifically, daytime AERONET AOD data are used for model training and testing. After training, nighttime MWR BT measurements are input into the model to generate nighttime AOD estimates. These estimates are then compared with nighttime AERONET lunar AOD measurements for validation.*

35. -> For me here there is not enough physical explanation how the AOD impact the BT and therefore how the BTs measured/retrieved with MWR can be used as proxy to retrieve AOD. Just say "machine learning" and explain inputs/output of a black box is not enough.

**Reply:** Thanks for your insightful concerns. Since the machine learning technique

does not directly reflect the physical relationship between aerosol loading and microwave radiances, we further verify the theoretical basis of our technique by analyzing the observed temperature and RH profiles under various AOD levels and using WRF-Chem combined with MonoRTM simulations.

Our simulation results (Figures 12 and 13) show that in the K band, BT increases with AOD for all frequencies, both day and night. For example, at 22.23 GHz, BT ranges from 60 K to 80 K under clean conditions and from 80 K to 130 K under polluted conditions, with statistically significant differences (Figures 12a and 13a). This trend is consistent across other K-band frequencies (Figures 12b-d and 13b-d) and is likely due to the hygroscopic growth of aerosols and their impact on water vapor, which enhances BT in the K band (Liu et al., 2014; Xie et al., 2013).

In contrast, the V band shows a decrease in BT with increasing AOD, observed in both daytime and nighttime (Figures 12e-h and 13e-h). For instance, at 51.76 GHz, BT ranges from 264 K to 270 K under clean conditions and from 262 K to 265 K under polluted conditions, with a statistically significant change ($p \leqslant 0.1$ by t-test).

This is attributed to the cooling effect of aerosols and the sensitivity of the V band to atmospheric temperature changes due to the oxygen absorption band (Van Leeuwen et al., 2001).

[Figure]

**Figure R9.** The boxplots of relationship between BT and AOD at 550 nm when fixing the surface temperature at 270-275 K from 00:00 UTC 18 December 2016 to 00:00 UTC 20 December 2016 in the WRF-Chem simulation. The frequencies of BT are (a)

22.23 GHz, (b) 23.03 GHz, (c) 25.00 GHz, (d) 28.00 GHz, (e) 51.76 GHz, (f) 53.34 GHz, (g) 54.94 GHz, and (h) 56.66 GHz during the daytime.

[Figure]

**Figure R10.** Similar to Figure R9, but for the nighttime.

We have revised the section concerning the physical explanation of how AOD impacts BT and how BTs measured/retrieved with MWR can be used as a proxy to retrieve AOD. The relevant revisions have been cited for reference (Lines 558-572 and Lines 573-586).

*Our simulation results, illustrated in Figure 12 and 13, indicate that for all frequencies in the K band, BT increases as AOD levels increase. This phenomenon exists in both the daytime and nighttime. Specifically, at 22.23 GHz, BT levels for clean conditions range from 60 K to 80 K, while for polluted conditions they range from 80 to 130 K, showing a statistically significant difference at both daytime and nighttime (Figure 12a & 13a). BT levels at other frequencies support this trend, indicating that BT tends to increase with AOD (Figure 12b-d & 13b-d). The increase of K band BT with AOD might be related to coherent changes of water vapor and aerosols, either due to aerosol absorption of water or meteorological conditions that affect both water vapor and aerosols. When AOD is higher, RH is higher, accompanied by more water vapor due to the hygroscopic growth effect of aerosols, as supported by previous analysis (Figure 11a & c). Since the K band includes the water vapor absorption line near 22.235 GHz, the BT in the K band is sensitive to water vapor, and thus the BT increases as AOD increases (Liu et al., 2014; Xie et al., 2013), further strengthening the theoretical foundation of the proposed approach.*

*In contrast to the observations in the K band, an analysis of the V band frequencies*

*reveals a consistent decrease in BT with the reduction of AOD levels, applicable to both diurnal and nocturnal periods (Figure 12e-h & 13e-h), which well corresponds to the cooling effect of aerosols. Notably, at a frequency of 51.76 GHz, the BT levels exhibit a range of 264 K to 270 K under pristine atmospheric conditions, whereas under polluted conditions, these levels are observed to be between 262 K and 265 K. Although the magnitude of this change is less pronounced than that observed in the K band, it passes the statistical significance ($p \leqslant 0.1$ by the t-test), indicating a reliable and measurable effect. The detailed physical interpretation as follows: due to the presence of the oxygen absorption band within the frequency range of the V band, it is highly sensitive to changes in atmospheric temperature (Van Leeuwen et al., 2001). Variations in AOD can influence the atmospheric temperature profile validated by observation and simulation (Figure 11b &d). Consequently, in cases where AOD is high, the BT in the V band decreases.*

36. L209: Subtly appears here another wavelength for the AOD: 550 nm. This is not a Photometer AOD (even if the AERONET product give it as computed value but not measured), this is the common (e. g. MODIS) satellite AOD channel... But you do not mention any satellite data. Is it the value given by the ECMWF ERA-5 model? Here also, be more precise and give a complete list of the parameter given by ERA-5 that you use.

**Reply:** Thank you for your insightful comments and for bringing the typo to our attention. The wavelengths in question are indeed 440 nm, 675 nm, 870 nm, and 1020 nm, which correspond to the output wavelengths of the RFR model. We have corrected this error in the manuscript and have cited the revision for reference (Lines 283-285).

*The relationship between AOD at 440 nm, 675 nm, 870 nm, and 1020 nm (the output wavelengths of the RFR model) and at the microwave band is enclosed in the random forest model.*

2.3 WRF-Chem simulations

**Reply:** Thank you for your comments. We have revised as suggested.

38. -> Do you really only needed to make 3 days and nights of WRF Chem simulation and then you have enough representative behavior to estimate AOD impact on the BTs in order to retrieve AOD from the MWR BT? If yes, this needs to be justified. I am afraid that you only have hibernal conditions since you are only simulating 3 days in December.

**Reply:** Thank you for your helpful suggestions. Actually, we have conducted two simulations. The first one runs from 00:00 UTC on 17 December 2016 to 00:00 UTC on 20 December 2016 (a 72-hour period). The another set of parallel experiments lasts from 00:00 UTC on 3 December 2016 to 00:00 UTC on 5 December 2016 (a 48-hour period) with the same settings. The first day of both sets of experiments is used for model stabilization, and the subsequent days are utilized for analysis. The selection of these two simulation periods is predicated upon the occurrence of pronounced pollution episodes, which endow these intervals with a heightened capacity to elucidate the impact of aerosols on meteorological fields and the concomitant microwave BTs.

Moreover, we also conducted a set of similar WRF-Chem simulations during the boreal summer to enhance the representativeness of the simulations, with specific simulation times from 00:00 UTC on 5 July 2016 to 00:00 UTC on 8 July 2016 (a 72-hour period). The selection of this simulation time period is also based on the serious pollution events that occurred in the Beijing area during this period. The findings also reveal that higher aerosol concentration levels have a negative effect on ST (Figure R11b & e), particularly during the daytime (Figure R11b), while positively influencing GDLR (Figure R11c & f), which is consistent with the MonoRTM calculations and two WRF-Chem simulation for the wintertime. Moreover, in the summertime WRF-Chem simulation, the relationship between AOD and BT remains consistent. Specifically, in the K band, BT exhibits an increasing trend with the rise in

AOD levels across all frequencies. Conversely, in the V band, a decrease in AOD levels consistently leads to a decline in BT. This pattern holds true for both diurnal and nocturnal periods, as illustrated in Figures R12 and R13 (Figure S3-S4 in the revised manuscript).

[Figure]

**Figure R11.** The difference of (a, d) aerosol optical depth (AOD, unitless), (b, e) surface temperature (ST, K), and (c, f) ground downward longwave radiation (GDLR, W/m²) between summertime EXP_AER and EXP_NOAER experiments (EXP_AER-EXP_NOAER) during the (a-c) daytime and (d-f) nighttime. The black circle indicates the MWR, and the dark grey square indicates the photometer. The daytime corresponds to the period from 22:00 UTC 7 July 2016 to 10:00 UTC 7 July 2016. The nighttime corresponds to the period from 10:00 UTC 7 July 2016 to 22:00 UTC 7 July 2016.

[Figure]

**Figure R12.** The boxplots of relationship between BT and AOD at 550 nm when fixing the surface temperature at 295-300 K from 00:00 UTC on 5 July 2016 to 00:00 UTC on 8 July 2016 in the WRF-Chem simulation. The frequencies of BT are (a) 22.23 GHz, (b) 23.03 GHz, (c) 25.00 GHz, (d) 28.00 GHz, (e) 51.76 GHz, (f) 53.34 GHz, (g) 54.94 GHz, and (h) 56.66 GHz during the daytime.

[Figure]

**Figure R13.** Similar to Figure R12, but for the nighttime.

We added the supplementary explanation in the revised manuscript and cited it here (Lines 333-336).

*The choice of these simulation periods is based on the presence of significant pollution events, which provide a robust basis for examining the influence of aerosols on meteorological fields and the associated microwave BTs.*

We also added the explanation about the simulation in the summer in the data and

method section and result section and cited them here (Lines 331-333 and Lines 610-614).

*Additionally, to augment the representativeness of our results, analogous WRF-Chem simulations were executed during the boreal summer from 00:00 UTC on 5 July 2016 to 00:00 UTC on 8 July 2016 (a 72-hour period).*

*Additionally, to augment the representativeness of our results, analogous WRF-Chem simulations were executed during the boreal summer. Specifically, these simulations were conducted from 00:00 UTC on 5 July 2016 to 00:00 UTC on 8 July 2016, covering a 72-hour duration, and they also yielded consistent conclusions (Figure S3-S5).*

39. Figure 3 -> Make an additional map with a zoom on the region of Beijing and make crosses for the 3 sites (AERONET station, MWR station, radiosonde station).

**Reply:** Thank you for your helpful suggestions. We have made an additional map with a zoom on the region of Beijing and make crosses for the 3 sites (Figure 2b in the revised manuscript). We also cited this figure here for convenience (Figure R14).

[Figure]

**Figure R14.** *(a) Simulation domains of the WRF-Chem experiments. (b) Left panel: the locations of observation sites in this study. The black circle indicates the MWR,*

*the dark grey square indicates the photometer, and the light triangle indicates the sounding. Right panel: the MWR used in this study is located in domain 3. This domain has a spatial resolution of 10 km. The MP-3000A MWR by Radiometrics is deployed at the Beijing Nanjiao Meteorological Observatory (39.80°N, 116.47°E) in China for brightness temperature (BT) measurements.*

**3. Results**

**3.1 Model fitting and validation**

L294-299 + Figure 4

40. -> I still do not understand why you do not consider reference wavelength 500 nm.

**Reply:** Thanks for your helpful suggestions and you are right. We have shown the results at 500 nm in the revised version and mainly discuss the model's performance at this wavelength. We also cited the revised statements here for convenience (Lines 383-389).

*The retrieval model has great fitting performance, as shown by Figure 4. The model fitting reaches correlation coefficients of 0.98 for the 440 nm, 500 nm, 675 nm, 870 nm, and 1020 nm, respectively, albeit with a minor systematic low bias for high AOD scenarios, which is similar to MODIS AOD products (Levy et al., 2013). Due to the consistent model performance in all wavelengths (Figure 4), we will focus on results at 500 nm in the following discussions since this is typically the reference wavelength for satellite remote sensing (Levy et al., 2013).*

41. -> Please explain better what is the train set: When, what (which odel), how, etc... and how it differs to the test set

**Reply:** Thank you for your helpful suggestions. We have added the explanation of the train set and how it differs to the test set. We cited it here for reference (Lines 274-276 and Lines 394-396).

*To ensure the representativeness of the sampling, we have partitioned the data such*

*that the 3/4 of the data in each month are designated as the training set, while the remaining 1/4 serves as the testing set.*

*The performance in 500 nm of the test set (R = 0.96, RMSE = 0.08, and MAPE = 0.11) is slightly inferior to the train set (R = 0.98, RMSE = 0.07, and MAPE = 0.10) regarding the statistical metrics (Figure 5a).*

42. L298-299: "we will focus on results 299 at 440 nm in the following discussions" -> should be better 500 nm for AOD

**Reply:** Thanks for your helpful suggestions and you are right. We have shown the results at 500 nm in the revised version and mainly discuss the model's performance at this wavelength. We also cited the revised statements here for convenience (Lines 386-389).

*Due to the consistent model performance in all wavelengths (Figure 4), we will focus on results at 500 nm in the following discussions since this is typically the reference wavelength for satellite remote sensing (Levy et al., 2013).*

43. Figure 4 + Figure 5: Is in y ("prediction") the MWR inverted product? If yes, write "MWR (prediction)" in y

**Reply:** Thank you for your helpful suggestions. You are right that this is the MWR inverted product and we have revised as suggested.

44. L305-306: "existing shortwave-based algorithms (Levy et al., 2013)." -> This is a satellite validation explain it, if not, one can believe that it is also a MWR or model validation

**Reply:** Thanks for your insightful suggestion and you are right. We have modified the explanation and cited it here for convenience (Lines 397-399).

*The accuracy of this estimation is similar to existing shortwave-based algorithms based on the satellite sensor such as the MODIS aerosol products (Levy et al., 2013).*

45. L314-315: reference to Figure 6: Explain in the text what is the AOD difference

**Reply:** Thanks for your helpful suggestions. We have explained the AOD differences between the MWR and the photometer. We also cited the explanation here for convenience (Lines 408-412).

*Moreover, the MWR tends to underestimate AOD during both daytime and nighttime, particularly at shorter wavelengths. As the wavelength increases, this underestimation diminishes, and the MWR retrieved AOD align more closely with AERONET observations (Figure 6). This trend is observed for both fine-mode and coarse-mode aerosols (Figure 6).*

L316-337 + figures 7 and 8

46. -> Again on the figures, if MWR then write "MWR Prediction" instead of only "Prediction"

**Reply:** Thank you for your helpful suggestions. You are right that this is the MWR inverted product and we have revised as suggested.

47. -> Here also explain the differences between the training set and the test set: When, how, etc...

**Reply:** Thank you for your helpful suggestions. We have added the explanation of the train set and how it differs to the test set. We cited it here for reference (Lines 413-416 and Lines 418-425).

*For retrieving vertical temperatures profiles, similarly to the AOD, we also partitioned the data such that the 3/4 of the data in each month are designated as the training set, while the remaining 1/4 serves as the testing set.*

*In detail, R is generally above 0.98 and all of the RMSEs are around 1.0 K in the training set (Figure 7). Similarly, the model's performance on the test set is somewhat lower compared to the training set, but remains satisfactory overall. Specifically, R is above 0.95 and all of the RMSEs are around 1.8 K the test set (Figure 8), comparable to previous studies using MWR to retrieval temperature profiles with an optimal*

*estimation method (Cimini et al., 2006).*

3.2 The diurnal cycle

48. -> Precise again from when to when is this study? Same as test (Dec 2019 - Oct 2020) If yes: Is it representative of this region? It was COVID Lockdown time in urban area, many studies have shown a special Aerosol load in this time. You may consider a longer (many years) time range to make a more relevant study.

**Reply:** Thank you for your helpful suggestions. The analysis period is indeed from December 2019 to October 2020, determined by the availability of data. We recognize that this period may not fully represent typical conditions in the region, particularly considering the potential impacts of the COVID-19 lockdown (Lv et al., 2020; Sulaymon et al., 2021). However, the influence of COVID-19 was primarily limited to the period from January 2020 to May 2020. By April 2020, Beijing had largely recovered from the pandemic, and industrial and other anthropogenic pollution sources returned to normal levels (Liu et al., 2022; Tao et al., 2021). Data availability constraints restricted us to using MWR data only within this timeframe. We have supplemented the expression accordingly and cited it here (Lines 440-447).

*We further examine the day-night differences in the AOD retrieved by MWR and compare them to those revealed by surface photometer. It should be noted that the analysis period in the following section remains from December 2019 to October 2020, contingent upon the availability of data. We acknowledge that the analysis period may not fully represent typical regional conditions due to COVID-19 (Lv et al., 2020; Sulaymon et al., 2021). However, the impact of COVID was mainly confined to January–March 2020. By April 2020, Beijing had largely recovered, with industrial and anthropogenic pollution sources returning to normal (Liu et al., 2022; Tao et al., 2021).*

-> Also: I suggest you to slice the study in 3 months groups: Summer: JUN-JUL-AUG; Spring: MAR-APR-MAY, Winter: DEC-JAN-FEB and fall: SEP-OCT-NOV, this is surely interesting to look if the mainstream 24 hours evolution

is the same in each season.

**Reply:** Thank you for your valuable suggestions. Following your advice, we have divided the study into four seasonal groups and analyzed the diurnal cycle of AOD. Our results show that the diurnal cycle of AOD is consistent across all seasons, with the most pronounced diurnal difference occurring in summer (Figure S1 in the revised manuscript). This significant diurnal variation in summer may be attributed to the intense solar radiation and high humidity levels, which are conducive to aerosol hygroscopic growth and thereby enhance aerosol extinction. We have incorporated the corresponding figure and discussion into the revised manuscript and cited them accordingly (Lines 480-491).

*We have further divided the results into four seasons and validated that the diurnal cycle of AOD is consistent across all seasons, with the most pronounced diurnal difference occurring in summer (Figure S1). It is noted that the lunar AOD is not available for JJA, which further underscores the supplementary role of MWR-predicted AOD in complementing lunar AOD measurements. The seasonal variation of AOD diurnal cycle agrees with previous studies derived from downscaling reanalysis datasets (Wang et al., 2025). The more significant diurnal difference in AOD during summer can be attributed to two primary factors. Firstly, the intense solar radiation and high temperatures prevalent in summer significantly promote the formation of aerosol particles through the process of gas-to-particle conversion (Chen et al., 2023a). Secondly, the high humidity levels in summer facilitate aerosol hygroscopic growth, which enhances aerosol extinction (Chen et al., 2023b; Lv et al., 2017).*

[Figure]

***Figure R15.*** *The diurnal cycle of seasonal mean MWR AOD and photometer AOD at 500 n including (a-c) MAM, (d-f) JJA, (g-i) SO, and (j-l) DJF. (a, d, g, j) The boxplot of hourly MWR AOD (red boxplots) and photometer AOD (blue boxplots). The small dots represent outliers greater than $q_{75} + 1.5(q_{75} - q_{25})$ or less than $q_{25} - 1.5(q_{75} - q_{25})$, where $q_{75}$ and $q_{25}$ correspond to 75th and 25th percentile. (b, e, h, k) The time series of mean AOD (solid lines) and median AOD (dashed lines) of MWR AOD (red lines) and photometer AOD (blue lines). (c, f, i, l) The boxplot of daytime and nighttime AOD. Blue boxes correspond to MWR data, and red boxes correspond to photometer data.*

49. -> And again: The reference AOD wavelength is 500 nm. Therefore if you want to show only one wavelength, then 500 nm please.

**Reply:** Thanks for your helpful suggestions and you are right. We have shown the results at 500 nm in the revised version and mainly discuss the model's performance at this wavelength. We also cited the updated Figure here for convenience.

[Figure]

***Figure R16.*** *The diurnal cycle of MWR AOD and photometer AOD at 500 nm. (a) The boxplot of hourly MWR AOD (red boxplots) and photometer AOD (blue boxplots). The small dots represent outliers greater than $q_{75} + 1.5(q_{75} - q_{25})$ or less than $q_{25} - 1.5(q_{75} - q_{25})$, where $q_{75}$ and $q_{25}$ correspond to 75th and 25th percentile. (b) The time series of mean AOD (solid lines) and median AOD (dashed lines) of MWR AOD (red lines) and photometer AOD (blue lines). (c) The boxplot of daytime and nighttime AOD. Blue boxes correspond to MWR data, and red boxes correspond to photometer data.*

50. L351-352 The decrease in AOD and the low number of measurements (photometer night) has no correlation: We can have very few measurements but all of them high AOD

**Reply:** Thanks for pointing it out and you are right. We have revised this statement and cited it here (Lines 458-460).

*This decrease may be attributed to the higher relative humidity near 23:00 and the corresponding aerosol scavenging effect, but further investigation is needed.*

3.3 Physical interpret

51. Since the machine learning technique does not necessarily represent the physical relationship between aerosol loading and microwave radiances -> This is why YOU MUST HAVE HAD WRITE A COMPLETE PART EXPLAINING HOW YOUR METHOD MAKE THE BRIDGE BETWEEN AEROSOL AND MW RADIANCES

**Reply:** Thank you for your astute observation. The complete part of explanation is in section 3.3. We have incorporated a statement to emphasize the crucial role of the simulation section in bridging the gap between aerosol loadings and microwave radiances. We also cite it here (Lines 512-513).

*The simulation is designed to establish a connection between aerosol loadings and microwave radiances.*

52. L403-408 and Figure 11: The variability of the temperature is so large, that the vertical profile pictures (a,d,b,e) have not relevance. (c) and (f) on the other side, are very interesting and show a large disagreement between WRF and observations, what should be commented and analyzed.

**Reply:** Thank you for pointing this out. We have noticed that there may be a significant discrepancy between WRF simulation results and observations with regards to the range, which can be partly attributed to model uncertainty. However, the overall patterns remain similar and the figure c and f do not significantly differ. For instance, in low-level polluted scenarios, the observed BT at 22.23 GHz ranges

from 42 K to 102 K, while the simulated range is narrower, from 55 K to 84 K. Despite the differences in the range, the overall patterns of BT between the WRF simulations and the observations are largely consistent. Figures c and f, in particular, exhibit minimal significant differences. For example, in low-level polluted scenarios, the observed BT at 22.23 GHz varies from 42 K to 102 K, whereas the simulated range is narrower, ranging from 55 K to 84 K.

Furthermore, although the absolute BT profiles may differ slightly, both the observational data and the WRF simulations reveal similar trends in BT as a function of AOD. This suggests that despite the range discrepancies, the fundamental relationships between BT and AOD are consistent between observation and simulation.

We also cited the supplementary explanation in the revised manuscript and cited it here (Lines 542-546).

*Although there might be a significant discrepancy of BT between WRF simulation results and observations with regards to the range, the trend and overall pattern is quite similar, revealing the similar trends in BT as a function of AOD (Figure 11f). This suggests that despite the range discrepancies, the fundamental relationships between BT and AOD are consistent between observation and simulation.*

53. L418-420: "The above observational evidence indicates that MWR estimate AOD by detecting the temperature and humidity profile differences caused by the presence of aerosols, further verifying the theoretical basis of our technique." -> 1) I do not agree maybe you have some season (winter) where Temperature are lower and aerosol different from other seasons are. Therefore you should slice your test analysis and graphics with seasons: JJA, SON, DJF.

**Reply:** Thank you for your valuable feedback. We have conducted a detailed seasonal analysis and corresponding graphical representation, and our findings align with the original conclusions. For the temperature profiles, we observe that higher AOD is associated with lower temperatures in the upper atmosphere, while the converse is

true (Figure 11a). In the low-level atmosphere, the temperature response to AOD is more complex, initially increasing with AOD before decreasing as AOD continues to rise. This variability can attributed to differences in aerosol type and optical properties across seasons (Che et al., 2024; Mahowald et al., 2011).

Regarding the RH vertical profiles, RH consistently increases with AOD across all pressure levels (Figure 11b), likely due to the hygroscopic growth of aerosols, which enhances AOD (Quan et al., 2018). Additionally, BTs at 22.23 GHz, calculated using the vertical profiles in MonoRTM, show a tendency to increase with AOD (Figure 11c).

These observational results suggest that MWRs may estimate AOD by detecting the differences in temperature and humidity profiles induced by the presence of aerosols. This insight motivates us to undertake further simulations to validate the theoretical underpinnings of our technique.

We also cited the figure and corresponding discussion here for reference (Lines 537-539).

*We have also conducted a detailed seasonal analysis and found similar responses in temperature, RH, and BT to AOD, with minor differences likely attributable to variations in aerosol types (Figure S2).*

[Figure]

***Figure R17.*** *(a-b, d-e, g-h, j-k) The seasonal observational vertical profiles of temperature (Temp, unit: K) and relative humidity (RH, unit: %) under various AOD levels at 550 nm. The cyan, orange, and red solid lines correspond to low-level polluted scenarios (AOD<0.2), mid-level polluted scenarios (0.2<AOD<0.5), and high-level polluted scenarios (AOD>0.5). (c, f, i, l) Their corresponding brightness temperature (BT, unit: K) at 22.23 GHz calculated by MonoRTM. These seasons include (a-c) MAM, (d-f) JJA, (g-i) SO, and (j-l) DJF. The shadings represent the spread of samples with one standard deviation. All differences have passed the significance test of p-value ≤0.01 by Student's t-test.*

54. 2) Just saying that "Aerosol influence the BT observed with MWR" is a very poor description of your algorithm, far away for the expectations of a scientific publication about a new and innovative technique.

**Reply:** Thanks for your concerns and you are right. This conclusion may seem somewhat arbitrary without solid evidence. However, our sensitivity analysis and model experiments demonstrate that microwave BT significantly responds to changes in aerosol load given the nearly same surface temperature. The impact of aerosols on microwave radiative transfer is highly complex, involving multiple processes such as aerosol scattering and absorption, changes in surface temperature and temperature/humidity profiles due to aerosol radiative and hygroscopic effects, and the nonlinear relationship between aerosol properties in the microwave and visible spectra. Additionally, both aerosols and water vapor may respond similarly to meteorological conditions. Given these complexities, constructing a physical retrieval model or a direct statistical relationship is nearly impossible, which is why we employed a machine learning-based technique.

The importance score of the random forest model, as shown in Figure 3 of the revised manuscript, confirms that the model reflects the physical relationship between BT and AOD. The significant contribution of BT from different channels as predictors underscores this relationship.

Additionally, there is a wide variety of research using the machine learning to retrieve aerosol properties, including aerosol optical depth (Chen et al., 2021; Lary et al., 2009; Logothetis et al., 2024), single scattering albedo (Dong et al., 2023; Wang et al., 2025), aerosol size (Rudiger et al., 2017), and aerosol types (Chen et al., 2021; Choi et al., 2021). All of these demonstrate the wide applicability and scientific basis of applying machine learning in estimating aerosol parameters.

We thus modify the statement and cited it here (Lines 546-554).

*The above observational evidence might indicate that MWR estimate AOD by detecting the temperature and humidity profile differences caused by the presence of aerosols, but the impact of aerosols on microwave radiative transfer is highly complex, involving multiple processes such as aerosol scattering and absorption, changes in surface temperature and temperature/humidity profiles due to aerosol radiative and*

*hygroscopic effects, and the nonlinear relationship between aerosol properties in the microwave and visible spectra. The above-mentioned complexities inspire us to conduct further simulation to verify the theoretical basis of our technique.*

55. L421-441 + Figures 12&13 -> All this study has no relevance if you first do not prove that aerosol load is not correlated to other events influencing the temperature statistic, like for instance the season. In many cities, in winter because of heating, the aerosol load is higher and the temperature of the atmosphere is lower, not because of aerosol cooling, but just because it is winter, the temperature is lower. On the other side, in some regions, in summer there are because of dryness more sand/earth aerosols and the temperature of the atmosphere is higher, not because of aerosol heating effect but because of summer. Therefore, making a one-year statistic and saying "the temperature changes: T atmo -> BT change, it is because of the aerosols" is not a serious atmospheric science argumentation.

**Reply:** Thank you for your insightful observation. Indeed, temperature plays a crucial role in determining microwave brightness temperatures (BT). To isolate the impact of AOD on BT, we have fixed the surface temperature at 270–275 K in our analysis. This specific range selection of surface temperature helps to minimize the influence of temperature variability on BT, thereby allowing the effects of AOD to be more prominent to our best. We have also highlighted this important detail in the revised manuscript and cited it accordingly (Lines 555-559).

*Furthermore, to isolate the impact of AOD on BT, we have fixed the surface temperature between 270 K and 275 K in our analysis. The selection of this specific surface temperature range effectively minimizes the influence of temperature variability on BT. Our simulation results, illustrated in Figure 12 and 13, indicate that for all frequencies in the K band, BT increases as AOD levels increase.*

56. L442-449: This experiment embitters the poor argumentation that I was cruising above. But nevertheless it is not enough. You have to deliver some keys how your algorithm is working: Give equation with the different BT that are used, and

explaining how you extracts. Presenting your retrieval algorithm, and showing tests to validate the different steps (in-between physical variables) computed by the algorithm.

**Reply:** Thank you for your affirmation, and I apologize for any confusion caused. The experiment comparing scenarios with aerosol loadings (EXP_AER) and without aerosol loadings (EXP_NOAER) is specifically designed to enhance our understanding of the impact of aerosol loading on longwave radiation, particularly its spatial distribution. Consequently, this simulation does not involve the use of brightness temperatures (BTs) since this is not computed within WRF. We have provided additional explanations in the revised manuscript and have cited the relevant sections here (Lines 600-602).

*This comparison is specifically designed to examine the impact of aerosol loading on longwave radiation, particularly its spatial distribution. As such, no BT information is generated or output in this comparison experiment.*

4. Conclusions and discussions

57. -> Please mention also in the conclusion some statistical values about the agreements between model results (WRF-Chem) and observations (MRW and/or AERONET) the same for the validation with comparison AERONET/MWR

**Reply:** Thanks for your helpful suggestions. We have mentioned some statistical values about the agreements between model results. We also cited it here (Lines 617-622 and Lines 636-640).

*By establishing a strong correlation (R = 0.96, RMSE = 0.11, and MAPE = 0.11 in the daytime test set) between the photometer AOD and multiple BTs derived from the MWR at the Beijing Nanjiao Meteorological Observatory using a machine learning algorithm, we were able to accurately retrieve nighttime AOD (R = 0.91, RMSE = 0.14, and MAPE = 0.28) and vertical temperature profiles (R > 0.95 for all levels and RMSE < 2.20 K for all levels).*

*Simulation further indicated a consistent and mostly linear increase in BTs in the K band (increasing from ~70 K to ~105 K at 22.23 GHz) and decrease in BTs in the V band (decreasing from ~265 K to ~257 K at 51.76 GHz) with AOD (550 nm, the wavelength of WRF-Chem simulated AOD) across all time periods.*

58. -> Please concerning the analysis of AOD, fine and coarse mode, specify that it is specific to one site (Beijing Nanjiao Meteorological Observatory in China) and precise which datetime range? and the conditions during the measurement date range (season, polluted/ not polluted / hot, wet, warm, etc...)

**Reply:** Thanks for your insightful suggestions. We have specified the site and time range of the AOD analysis and included the measurement conditions in the revised manuscript. We also cited them here (Lines 649-652).

*It is important to note that the analysis of AOD is specifically conducted for the Beijing Nanjiao Meteorological Observatory in China, covering the period from December 2019 to October 2020. This timeframe encompasses various climate and pollution conditions and is contingent upon the availability of data.*

FIGURES:

59. Figure 1: What do you do with the profile measurements with the radiosonde mentioned at Line 172?

**Reply:** Thanks for your suggestions. The collocation process between radiosonde and MWR involves identifying the temporally nearest valid BT measurement and subsequently inputting these BT values into the RFR model to generate the MWR-based vertical temperature profile prediction. The radiosonde vertical temperature profiles are then vertically interpolated to the standard pressure levels (100 hPa, 200 hPa, 500 hPa, 700 hPa, 850 hPa, and 1000 hPa) using a linear interpolation method, allowing for direct comparison with the MWR-based vertical temperature profile prediction. We also clarified it in the section 2.1 datasets section, modified the Figure 1, and cited it here (Lines 212-222).

*To further assess the accuracy of the model in predicting vertical temperature profiles, we utilized the collocated sounding data obtained from Beijing Meteorological Station (station ID: 54511) during the corresponding time frame. The collocation process involves identifying the temporally nearest valid BT measurement and subsequently inputting this BT value into the model to generate the MWR-based vertical temperature profile prediction. The radiosonde temperature profiles are then vertically interpolated to the standard pressure levels (100 hPa, 200 hPa, 500 hPa, 700 hPa, 850 hPa, and 1000 hPa) using a linear interpolation method, allowing for direct comparison with the MWR-based temperature profile prediction. These sounding data were collected twice daily respectively at 00:00 and 12:00 UTC from December 2019 to October 2020.*

[Figure]

**Figure R18.** *The flowchart for clear sky nighttime AOD and vertical temperature profiles retrieval algorithm.*

60. Figure 2 (Caption): "MWR Channel" -> "MWR Channels"

**Reply:** Thanks for your suggestions. We have revised as suggested.

61. Figure 4,5: Why "prediction" and not "MWR"

**Reply:** Thanks for your suggestions. We have revised as suggested.

**Reply:** Thanks for your helpful suggestions and you are right. We have shown the results at 500 nm in the revised version and mainly discuss the model's performance at this wavelength. We also cited the updated Figure here for convenience.

[Figure]

***Figure R19.*** *Density scatterplots of 500 nm AOD in the test set of MWR and the photometer with (a) daytime, and (b) nighttime. The dashed dark gray line represents the 1:1 line, and the black solid line represents the linear regression line. Note that the daytime corresponds to 6:00 am to 6:00 pm for the local time (UTC+8), and nighttime corresponds to the remaining time.*

63. Figure 10: Same comment as Figure 5: Why focus on 440 nm? 500 nm is the standard value

**Reply:** Thanks for your helpful suggestions and you are right. We have shown the results at 500 nm in the revised version and mainly discuss the model's performance at this wavelength. We also cited the updated Figure here for convenience.

[Figure]

***Figure R20.*** *The diurnal cycle of MWR AOD and photometer AOD at 500 nm. (a) The boxplot of hourly MWR AOD (red boxplots) and photometer AOD (blue boxplots). The small dots represent outliers greater than $q_{75} + 1.5(q_{75} - q_{25})$ or less than $q_{25} - 1.5(q_{75} - q_{25})$, where $q_{75}$ and $q_{25}$ correspond to 75th and 25th percentile. (b) The time series of mean AOD (solid lines) and median AOD (dashed lines) of MWR AOD (red lines) and photometer AOD (blue lines). (c) The boxplot of daytime and nighttime AOD. Blue boxes correspond to MWR data, and red boxes correspond to photometer data.*

64. Figure 11: AOD at which wavelength?

**Reply**: Thanks for pointing it out. This is AOD at 550 nm. We have supplemented the caption in the revised manuscript accordingly.

65. Figure 12 & 13: Better do it with AOD at 500nm than 550 nm? 500 nm is the reference. 550 nm is only computed but not measured by AERONET.

**Reply**: Thank you for raising this point. WRF-Chem does not output AOD at 500 nm. Therefore, we selected 550 nm as the closest available alternative wavelength in the WRF-Chem output. This choice has been clarified in the revised manuscript and cited accordingly.

*Here, we utilize AOD at 550 nm instead of 500 nm because WRF-Chem does not simulate AOD at 500 nm. Thus, 550 nm was selected as the closest available alternative wavelength in the WRF-Chem output.*

66. Figure 14 (caption): Please write the hours HH:MM, example: 20:00 UTC (and not "2000 UTC")

**Reply:** Thanks for your helpful suggestions. We have revised as suggested.

67. Figure 14: Please on each map, make a cross where the AERONET station is and where the MWR station is.

**Reply:** Thanks for your helpful suggestions. We have made crosses as suggested and cited the revised figure here for reference.

[Figure]

**Figure R21.** *The difference of (a, d) aerosol optical depth (AOD), (b, e) surface temperature (ST), and (c, f) ground downward longwave radiation (GDLR) between EXP_AER and EXP_NOAER experiments (EXP_AER-EXP_NOAER) during the (a-c) daytime and (d-f) nighttime. The black circle indicates the MWR, and the dark grey square indicates the photometer. The daytime corresponds to the period from 22:00 UTC 18 December 2016 to 10:00 UTC 19 December 2016. The nighttime corresponds to the period from 10:00 UTC 19 December 2016 to 22:00 UTC 19 December 2016.*

**Reference**

Barreto, A., Roman, R., Cuevas, E., Berjon, A. J., Fernando Almansa, A., Toledano, C., Gonzalez, R., Hernandez, Y., Blarel, L., Goloub, P., Guirado, C., and Yela, M.: Assessment of nocturnal aerosol optical depth from lunar photometry at the Izana high mountain observatory, ATMOSPHERIC MEASUREMENT TECHNIQUES, 10, 3007-3019, 10.5194/amt-10-3007-2017, 2017.

Berkoff, T. A., Sorokin, M., Stone, T., Eck, T. F., Hoff, R., Welton, E., and Holben, B.: Nocturnal Aerosol Optical Depth Measurements with a Small-Aperture Automated Photometer Using the Moon as a Light Source, JAtOT, 28, 1297-1306, 10.1175/JTECH-D-10-05036.1, 2011.

Che, H. Z., Xia, X. G., Zhao, H. J., Li, L., Gui, K., Zheng, Y., Song, J. J., Qi, B., Zhu, J., Miao, Y. C., Wang, Y. Q., Wang, Z. L., Wang, H., Dubovik, O., Holben, B., Chen, H. B., Shi, G. Y., and Zhang, X. Y.: Aerosol optical and radiative properties and their environmental effects in China: A review, EARTH-SCIENCE REVIEWS, 248, 10.1016/j.earscirev.2023.104634, 2024.

Chen, X., Zheng, F., Guo, D., Wang, L., Zhao, L., Li, J., Li, L., Zhang, Y., Zhang, K., Xi, M., and Li, K.: Review of machine learning methods for aerosol quantitative remote sensing, National Remote Sensing Bulletin, 25, 2220-2233, 2021.

Choi, W., Lee, H., and Park, J.: A First Approach to Aerosol Classification Using Space-Borne Measurement Data: Machine Learning-Based Algorithm and Evaluation, REMOTE SENSING, 13, 10.3390/rs13040609, 2021.

Dong, Y., Li, J., Yan, X., Li, C., Jiang, Z., Xiong, C., Chang, L., Zhang, L., Ying, T., Zhang, Z., and Wang, M.: Retrieval of aerosol single scattering albedo using joint satellite and surface visibility measurements, REMOTE SENSING OF ENVIRONMENT, 294, 10.1016/j.rse.2023.113654, 2023.

Ge, J., Huang, J., Weng, F., and Sun, W.: Effects of dust storms on microwave radiation based on satellite observation and model simulation over the Taklamakan desert, ATMOSPHERIC CHEMISTRY AND PHYSICS, 8, 4903-4909, 10.5194/acp-8-4903-2008, 2008.

Hong, G., Yang, P., Weng, F. Z., and Liu, Q. H.: Microwave scattering properties of sand particles: Application to the simulation of microwave radiances over sandstorms, J. Quant. Spectros. Radiat. Transfer, 109, 684-702, 10.1016/j.jqsrt.2007.08.018, 2008.

Lary, D. J., Remer, L. A., MacNeill, D., Roscoe, B., and Paradise, S.: Machine Learning and Bias Correction of MODIS Aerosol Optical Depth, IEEE GEOSCIENCE AND REMOTE SENSING LETTERS, 6, 694-698, 10.1109/LGRS.2009.2023605, 2009.

Liu, D. W., Lv, C. C., Liu, K., Xie, Y., and Miao, J. G.: Retrieval Analysis of Atmospheric Water Vapor for K-Band Ground-Based Hyperspectral Microwave Radiometer, IEEE GEOSCIENCE AND REMOTE SENSING LETTERS, 11, 1835-1839, 10.1109/LGRS.2014.2311833, 2014.

Liu, G. Y., Li, J., Li, J., Yue, S., and Zhou, R. L.: Estimation of Nighttime Aerosol Optical Depths Using Atmospheric Infrared Sounder Longwave Radiances, GEOPHYSICAL RESEARCH LETTERS, 51, 10.1029/2023GL108120, 2024.

Liu, S., Yang, X. C., Duan, F. Z., and Zhao, W. J.: Changes in Air Quality and Drivers for the Heavy PM2.5 Pollution on the North China Plain Pre- to Post-COVID-19, INTERNATIONAL JOURNAL OF ENVIRONMENTAL RESEARCH AND PUBLIC HEALTH, 19, 10.3390/ijerph191912904, 2022.

Logothetis, S.-A., Salamalikis, V., and Kazantzidis, A.: A Machine Learning Approach to

Retrieving Aerosol Optical Depth Using Solar Radiation Measurements, REMOTE SENSING, 16, 10.3390/rs16071132, 2024.

Lv, Z. F., Wang, X. T., Deng, F. Y., Ying, Q., Archibald, A. T., Jones, R. L., Ding, Y., Cheng, Y., Fu, M. L., Liu, Y., Man, H. Y., Xue, Z. G., He, K. B., Hao, J. M., and Liu, H. A.: Source-Receptor Relationship Revealed by the Halted Traffic and Aggravated Haze in Beijing during the COVID-19 Lockdown, Environ Sci Technol, 54, 15660-15670, 10.1021/acs.est.0c04941, 2020.

Mahowald, N., Ward, D. S., Kloster, S., Flanner, M. G., Heald, C. L., Heavens, N. G., Hess, P. G., Lamarque, J. F., and Chuang, P. Y.: Aerosol Impacts on Climate and Biogeochemistry, in: ANNUAL REVIEW OF ENVIRONMENT AND RESOURCES, VOL 36, edited by: Gadgil, A., and Liverman, D. M., 45-74, 10.1146/annurev-environ-042009-094507, 2011.

Quan, J. N., Jiang, C. L., Xin, J. Y., Zhao, X. J., Jia, X. C., Liu, Q., Gao, Y., and Chen, D.: Evaluation of satellite aerosol retrievals with in situ aircraft and ground measurements: Contribution of relative humidity, ATMOSPHERIC RESEARCH, 212, 1-5, 10.1016/j.atmosres.2018.04.024, 2018.

Rudiger, J. J., Book, K., degrassie, J. S., Hammel, S., and Baker, B.: A machine learning approach for predicting atmospheric aerosol size distributions, LASER COMMUNICATION AND PROPAGATION THROUGH THE ATMOSPHERE AND OCEANS VI, 2017, WOS:000417336700019, 10.1117/12.2276717, 2017.

Sulaymon, I. D., Zhang, Y. X., Hopke, P. K., Hu, J. L., Zhang, Y., Li, L., Mei, X. D., Gong, K. J., Shi, Z. H., Zhao, B., and Zhao, F. X.: Persistent high PM2.5 pollution driven by unfavorable meteorological conditions during the COVID-19 lockdown period in the Beijing-Tianjin-Hebei region, China, Environ Res, 198, 10.1016/j.envres.2021.111186, 2021.

Tao, C. L., Wheiler, K., Yu, C., Cheng, B. D., and Diao, G.: Does the joint prevention and control regulation improve the air quality? A quasi-experiment in the Beijing economic belt during the COVID-19 pandemic, SUSTAINABLE CITIES AND SOCIETY, 75, 10.1016/j.scs.2021.103365, 2021.

Van Leeuwen, G. M. J., Hand, J. W., de Kamer, J. B., and Mizushina, S.: Temperature retrieval algorithm for brain temperature monitoring using microwave brightness temperatures, ELECTRONICS LETTERS, 37, 341-342, 10.1049/el:20010269, 2001.

Wang, Q., Li, S., Zhang, Z., Lin, X., Shuai, Y., Liu, X., and Lin, H.: Retrieving aerosol single scattering albedo from FY-3D observations combining machine learning with radiative transfer model, ATMOSPHERIC RESEARCH, 315, 10.1016/j.atmosres.2024.107884, 2025.

Xie, Y., Chen, J. X., Liu, D. W., Lv, C. C., Liu, K., and Miao, J. G.: DEVELOPMENT AND CALIBRATION OF A K-BAND GROUND-BASED HYPERSPECTRAL MICROWAVE RADIOMETER FOR WATER VAPOR MEASUREMENTS, PROGRESS IN ELECTROMAGNETICS RESEARCH-PIER, 140, 415-438, 10.2528/PIER13050704, 2013.

Zhang, L. L., Liu, M. J., He, W. Y., Xia, X. A., Yu, H. N., Li, S. X., Li, J.: Ground Passive Microwave Remote Sensing of Atmospheric Profiles Using WRF Simulations and Machine Learning Techniques, Journal of Meteorological Research, 10.1007/s13351-024-4004-2, 2024.